# CAR T Cell Nanosymbionts: Revealing the Boundless Potential of a New Dyad

**DOI:** 10.3390/ijms252313157

**Published:** 2024-12-07

**Authors:** Juan C. Baena, Lucy M. Pérez, Alejandro Toro-Pedroza, Toshio Kitawaki, Alexandre Loukanov

**Affiliations:** 1Division of Oncology, Department of Medicine, Fundación Valle del Lili, ICESI University, Carrera 98 No. 18-49, Cali 760032, Colombia; lucyperezlugo@outlook.com (L.M.P.); alejandro.toro1@u.icesi.edu.com (A.T.-P.); 2LiliCAR-T Group, Fundación Valle del Lili, ICESI University, Cali 760032, Colombia; 3Department of Hematology, Kyoto University Hospital, Kyoto 606-8507, Japan; kitawaki@kuhp.kyoto-u.ac.jp; 4Department of Chemistry and Materials Science, National Institute of Technology, Gunma College, Maebashi 371-8530, Japan; loukanov@gunma.kosen-ac.jp; 5Laboratory of Engineering Nanobiotechnology, University of Mining and Geology “St. Ivan Rilski”, 1700 Sofia, Bulgaria

**Keywords:** CAR T cell therapy, immunoengineering, nanotechnology, clinical translation, nanoparticle, drug delivery, biomaterials, gene delivery, nano-immunotherapy, nanomachines, nanorobotics, living–nonliving integration, material-based biological regulation, cell-based therapy

## Abstract

Cancer treatment has traditionally focused on eliminating tumor cells but faces challenges such as resistance and toxicity. A promising direction involves targeting the tumor microenvironment using CAR T cell immunotherapy, which has shown potential for treating relapsed and refractory cancers but is limited by high costs, resistance, and toxicity, especially in solid tumors. The integration of nanotechnology into ICAM cell therapy, a concept we have named “CAR T nanosymbiosis”, offers new opportunities to overcome these challenges. Nanomaterials can enhance CAR T cell delivery, manufacturing, activity modulation, and targeting of the tumor microenvironment, providing better control and precision. This approach aims to improve the efficacy of CAR T cells against solid tumors, reduce associated toxicities, and ultimately enhance patient outcomes. Several studies have shown promising results, and developing this therapy further is essential for increasing its accessibility and effectiveness. Our “addition by subtraction model” synthesizes these multifaceted elements into a unified strategy to advance cancer treatment paradigms.

## 1. Introduction

Over the past century, cancer treatment has been focused on identifying and eliminating malignant cells, while often neglecting the surrounding environment. While this approach has yielded some results, these are not consistently promising, as over time, resistance mechanisms often undermine initial successes, particularly in advanced stages. Treatments such as immune checkpoint inhibitors, on-target therapy, and conventional cytotoxic chemotherapy are not exceptions to this reality [1]. Broadening our focus and targeting the tumor microenvironment could represent a significant advancement in cancer treatment [2]. Adoptive cell therapy, specifically chimeric antigen receptor T (CAR T) cell therapy, is revolutionizing the way relapsed and refractory neoplasms are treated. The CAR T cell approach is interesting and innovative due to its disruptive mechanism of action: taking advantage of T lymphocytes’ direct cytotoxicity using a modified membrane receptor with much higher binding affinity and specific immunoglobulin capabilities without the natural filtering constraints of the major histocompatibility complex to destroy cancer cells (see Figure 1a). However, such a boundary-breaking model of therapeutics must deal with several challenges [3].

The emergence of CAR T cell therapy has paved the way for a new era in cancer therapy, especially for the treatment of hematological malignancies [4]. Kymriah and Yescarta have been commercially available since 2017 and 2018, respectively, and the U.S. Food and Drug Administration (FDA) has approved six CAR T cell therapies to date [5] (see Figure 1b). Despite these advances, CAR T cell therapy is far from perfect [6]. Following its infusion, several side effects, such as cytokine release syndrome (CRS), neurological toxicity (ICANS), and on-target off-tumor effects (OTOT) can emerge. Resistance and recurrence of tumors after CAR T treatment occur in 30–50% of patients, and solid tumors are a forbidden field for therapy due to their overall response rate of 9%. Lastly but of no lesser importance, the prohibitive cost of CAR T cell manufacturing and administration restricts the widespread use of this treatment [7,8] (see Figure 1c).

Nanotechnology is a scientific field focused on the engineering and application of materials and devices with structures and properties at the nanoscale, typically ranging from 1 to 100 nanometers (nm). At this scale, materials often exhibit novel properties that differ significantly from their bulk counterparts. Nanomaterials are broadly categorized into inorganic, organic, and hybrid types. Inorganic nanoparticles include metals such as silver, gold, and platinum, as well as metal oxides such as iron oxide, while organic nanoparticles encompass natural and synthetic lipids, polymers, dendrimers, and other organic supramolecular assemblies [4]. Each type of nanoparticle varies in payload flexibility, surface charge, and abundance of chemical groups and possesses significant potential for conjugation to diverse biomolecules for active targeting. Additionally, nanoparticles exhibit unique physical and chemical properties such as size, structure, and geometry as well as electrical, magnetic, optical, and catalytic characteristics. These attributes, coupled with enhanced bioavailability and improved encapsulation efficiency, collectively influence their applications in theranostic approaches [9], enabling them to overcome biological barriers and enhance cancer treatment. Some of these nanoparticles have already been approved by the FDA for clinical use and play a crucial role in various facets of cancer theranostics, including diagnosis and therapies such as chemotherapy and immunotherapy. For example, medication based on nanoparticles is effectively used for targeted delivery of antigens to dendritic cells; controlled release of inhibitors, gene therapy vectors, antigens, and adjuvants into tumors; conjugation to T cells and natural killer cells for prolonged activation; transfection of dendritic cells to produce effective cancer vaccines; and modulation of immune cell subtypes [5]. Nanoparticles can carry multiple components simultaneously—such as tumor-associated antigens, adjuvants (immune stimulators), and targeting molecules—allowing the creation of multifunctional platforms. This combination approach can enhance the immune response in a more coordinated and efficient way, boosting both the innate and adaptive immune systems. The expansive potential of nanotechnology offers new opportunities for integrating nanoparticles into CAR T cell immunotherapy to address its current limitations [6] (see Figure 2). Recent advancements in nanotechnology have led to the emergence of nanomachines and nanorobots, increasingly utilized in medicine for surgical assistance, diagnostics, and drug delivery [10]. Typically, these smart nanodevices comprise an internal payload and an external shell designed for propulsion and precise targeting, representing a significant breakthrough in nanomedicine. Theranostic nanomachines [11] enable a higher level of targeted drug delivery, surpassing conventional therapies, due to their integrated therapeutic and imaging capabilities that advance personalized medicine. This innovation facilitates more accurate diagnosis, tailored treatments, and potential enhancements to CAR T immunotherapy for solid tumors. Designing effective nanomachines and nanorobotics involves ensuring biocompatibility with healthy tissues and navigating intelligently through the heterogeneous and diverse tumor microenvironments. This capability is crucial for targeting cancer cells based on specific biomarkers and environmental cues. Nanomachines respond to local microenvironmental stimuli such as temperature, light, X-rays, ultrasound, magnetic radiation, pH changes, ionic strength, etc., making them versatile tools for cancer theranostics and giving them promising applications in CAR T immunotherapy. They facilitate drug delivery with enhanced targeting and penetration, offer potential to overcome immunological and biological barriers, enable precise cancer cell recognition through active targeting strategies, and achieve improved controlled drug release with triggered systems. These capabilities underscore their impressive molecular-level performance and potential for synergistic therapies when integrated into combination therapy strategies for cancer treatment. In addition, advanced nanotechnology enables the possibility of developing personalized anticancer vaccines. Tumor antigens can be loaded onto nanoparticles tailored to the individual patient’s tumor profile. These personalized vaccines can trigger a specific immune response against the patient’s unique tumor antigens, improving treatment outcomes.

Currently, there is growing interest among clinicians and researchers in harnessing the advantages of nanotechnology for improved CAR immune cell manufacturing, CAR immune cell-assisted delivery, modulation of CAR T cell activity, and targeting the tumor microenvironment [7]. These approaches aim to establish a symbiotic relationship between chimeric antigen receptor T cell therapy and nanomedicine, termed “CAR T nanosymbionts”, designed to enhance the therapeutic capabilities of CAR T cell therapies for treating solid tumors. In the following sections, we will discuss the current biological challenges derived from the concept of CAR T cells as living drugs, alongside potential solutions offered by CAR T nanosymbionts. We will also discuss the obstacles of the ongoing CAR T cell manufacturing process and specific instances where CAR T nanosymbionts can be used to overcome these barriers. Finally, we will present our forward-looking perspective in these areas, including a theoretical model aimed at enhancing the flexibility and adaptability of the current standard.

## 2. Biological Challenges Arising from the Concept of CAR T Cells as Living Drugs and Potential Solutions Offered by CAR T Nanosymbionts

### 2.1. The Immune Synapse (IS)

The immune synapse (IS) is crucial for CAR T cell activation, triggering cytotoxic lymphocytes (CTLs) [12,13,14]. Unlike conventional T cells, CAR T cells form a disorganized IS configuration, using Lck/ZAP70 signaling to rapidly establish the CAR T cell IS (CAR-IS) within 5 min. This leads to intense signaling, quick detachment from target cells, and a “serial killer” pattern of cancer cell destruction [15,16] (see Figure 1a). CAR-IS shows higher expression of molecules such as Bcl2 and PEA-15 [17], which have antiapoptotic and antiproliferative properties [18]. Enhancing Fas-mediated apoptosis through agents such as histone deacetylase inhibitors (HDAC inhibitors or HDACi) can boost CAR T cell effects [19], but systemic HDACi use has challenges such as poor pharmacokinetics, low specificity, and drug resistance [20].

The nanocarriers and nanomachines can deliver these molecules, enhancing CAR-IS and overcoming CAR T cell resistance. To address this, Singleton and colleagues designed a nano-micelle system for glioma treatment composed of poloxamer 407 with a hydrophobic polypropylene glycol chain in the middle and two hydrophilic polyethylene glycol (PEG) molecules at both ends. This enhanced the delivery and efficacy of panobinostat, increasing its intracranial concentration and improving glioma response and survival in a murine model [21]. Several approaches using nanoparticles and its properties (passive targeting, ligand modification on its surface, light, ultrasound, external alternating magnetic field, redox potential, pH, and temperature responses) are being explored for delivering different HDACi such as vorinostat, quisinostat, and belinostat [22]. Birinapant works both as an antagonist of IAPs and a mimetic compound of the endogenous second mitochondria derived activator of caspases, effectively promoting apoptosis of cancer cells. The solubility of birinapant is pH-dependent, which has led researchers to create a specialized mixed micellar formulation known as PPB/MPP. This formulation combines birinapant with poly (ethylene glycol) and palmitic acid (PAL) to form PAL–PEG 4k–birinapant (PPB) and mixes this product with 3-(4,5-dimethylthiazol-2-yl)-2,5-diphenyl-tetrazolium bromide (MPP). Additionally, paclitaxel (PTX), a well-known anticancer drug, was loaded into the mixed micelles to form PTX-loaded PPB/MPP. PTX and birinapant entrapped in PPB/MPP micelles allows the rapid release of these drugs in environments with a pH of 5.0 and has been proven to enhance in vitro and in vivo cytotoxicity. Once released into the cytoplasm, PTX exerts antimicrotubular activity while birinapant inhibits IAPs leading to cell apoptosis of cancer cells with an excellent safety profile, even with several times the maximum tolerated dose for systemic infusions [23]. The effectiveness of CAR T cells depends on high tumor antigen density [24,25], but this is often variable. Modifiable factors such as co-stimulatory domains influence CAR T performance. Enhancing signal strength can involve altering co-stimulatory domains, adding activation motifs, using chimeric receptors, or modifying the hinge-transmembrane region [26,27]. Proper selection of these domains is key to improving CAR T cell therapies.

Nanoparticles, as vectors enveloped with cargo (or nanovectors), also offer a novel approach as carriers of mRNA or relatively large plasmids containing the genetic material necessary for these enhancements [8]. These nanoforms have been observed in some trials where in vivo generation of CAR T cells was selectively performed in human CD4+ lymphocytes [28]. Nanomachines represent an advanced technique for transferring specific mRNA in vivo, targeting different subtypes of lymphocytes through antibodies attached to the surfaces of nanoparticles, ensuring precise delivery of the genetic material. For example, virus-like nanoparticles can be engineered using bioorthogonal chemistry to exhibit prolonged blood circulation, reduced immunogenicity, efficient gene delivery to target cells, and safety for systemic gene therapy. These properties contribute to their remarkable therapeutic efficacy in treating certain cancers. In the case of breast cancer, low pH levels trigger their operation through structural conformational changes [29].

The immunological synapse, essential for CAR T cell function, relies on the precise interaction between the CAR’s antigen-binding domain and tumor cell epitopes. Currently, this recognition is mediated by the scFv fragment, which can suffer from misfolding, aggregation, and overstimulation of T cells, leading to early exhaustion. Replacing the scFv with a nanobody—a small, single-domain antibody derived from camelids—offers significant improvements. Nanobodies, due to their compact size, high stability, and reduced immunogenicity, create a more efficient and stable synapse. Their ability to avoid misfolding and aggregation ensures a more controlled activation of CAR T cells, preventing the overexpression of cytotoxic signals that could prematurely exhaust the T cells. By enhancing specificity and affinity, nanobodies improve antigen recognition, leading to a stronger and more durable synapse between the CAR T cells and tumor cells. Additionally, nanobodies can be engineered into modular structures that facilitate the redirection of universal CAR T cells to target various tumor antigens, further enhancing the precision and adaptability of the therapy. This combination of stability, specificity, and versatility helps optimize the immune synapse, improving both the efficacy and the longevity of CAR T cell responses against cancer [30].

#### Beyond Binding: The Immune Synapse as a Multi-Angle View

While tumor-associated antigens (TAA) are key to CAR T cell function, identifying alternative specific tumor antigens is difficult. Even when a candidate is found, many antigen-positive cells do not respond to CAR T cell reinfusion, implying that multiple pathways are involved in immune synapse function and CAR T cell cytotoxicity. In solid tumors, interferon-γ receptor (IFNγR) signaling is critical for cell adhesion after CAR T cell treatment. Disruptions in this signaling can impair immune synapse formation, reduce CAR T cell binding, and lead to resistance. This has been noted in glioblastoma as well as ovarian, lung, and pancreatic cancers, where ICAM1 expression is reduced when IFNγR signaling is impaired. Conversely, soluble IFNγ can increase ICAM-1 expression but may also trigger cytokine toxicity depending on the tumor type [31].

Drug delivery nanodevices have the potential to improve the performance of immune synapses and their interaction with the tumor microenvironment. Siriwon and colleagues achieved the modification of the A2a adenosine receptor (A2aR) inhibitory pathway using nanomachines. The A2aR pathway, which is triggered by increased adenosine levels caused by tissue damage and cellular stress, inhibits T cell receptor signaling and IFNγ production through elevated intracellular cyclic AMP adenosine levels that are increased in the tumor microenvironment [32]. They used previous data which demonstrated that several adenosine receptor blockers reduced tumor growth, metastasis, and neovascularization [33]. To leverage these insights, researchers developed SCH-58261-loaded, cross-linked multilamellar liposomes with maleimide functionalization on the surface of CAR T cells. This strategy was applied in models of ovarian cancer and chronic myelogenous leukemia. The approach did not impair the effector function of CAR T cells and utilized these cells as vehicles to enhance the colocalization of nanoparticles in sensitive tumor areas and prolonging tumor growth inhibition by targeting the A2a receptor pathway. Simultaneously, this method demonstrated the capacity to rejuvenate tumor-resident T cells in vivo, eliciting an adjuvant phenomenon [34].

The SELEX (Systematic Evolution of Ligands by Exponential Enrichment) technique is a robust method for identifying aptamers—short, single-stranded nucleic acids capable of binding to target cancer cells with high specificity and affinity, akin to antibodies. When integrated with drug delivery nanodevices, these screened aptamers can enhance immune synapses in solid tumors by improving the targeting, delivery, and efficacy of therapeutic agents. Nanocarriers functionalized with aptamers can deliver cytokines, immune checkpoint inhibitors, or cytotoxic drugs directly to the tumor microenvironment, facilitating targeted therapy of cancer cells. By binding to specific molecules on cancer cell membranes, aptamers identified through cancer biopsy can accumulate within solid tumors. Theoretically, CAR T cells can be engineered to recognize these aptamer-labeled cancer cells. This involves modifying CAR T cells to include a domain that interacts with the selected aptamer sequence. This approach broadens the potential range of antigens targeted by CAR T cells, including those that are not readily recognized by conventional antibodies. By delivering immune modulators directly to the tumor microenvironment, aptamer-functionalized nanocarriers enhance the formation and function of immune synapses between CAR T cells and cancer cells. This targeted approach holds promise for achieving a more effective anti-tumor immune response [35].

### 2.2. CAR T Exhaustion

T cell exhaustion occurs due to increased inhibitory signals from molecules such as PD-1, Tim-3, LAG-3, VISTA, CTLA-4, and TIGIT, affecting both tumor cells and T lymphocytes. It is common in over-differentiated CAR T cells, linking a naïve T cell phenotype to better function and clinical outcomes [36]. Factors such as prior chemotherapy and conditioning regimens such as lymphodepletion contribute to this exhaustion. Moreover, systemic administration of immune checkpoint inhibitors may exacerbate toxicity without effectively concentrating at the tumor site. Recent strategies have shown promise in mitigating CAR T cell exhaustion and reducing relapse rates through the use of bioengineered polymer matrices that serve as dynamic reservoirs from which CAR T cells are deployed [37,38]. These approaches employ biomaterials to enhance T cell migration rates observed in lymphoid tissues and incorporate an interleukin 15 superagonist and co-stimulatory domain antibodies (e.g., anti-CD3, anti-CD28, and anti-CD137) within the microsphere bilayers to pre-activate T cells, thereby enhancing their proliferation and migration capabilities. These systems have been evaluated in murine models of breast and ovarian cancer with incompletely resected or inoperable tumors and where migration and host lymphodepletion was unnecessary. Additionally, the use of bio-polymer scaffolds might extend to adjuvant therapies for completely resected tumors and in managing metastatic scenarios [37,39].

Nanocarriers [36], developed by Weidong Nie, can effectively deliver PD-L1 antibodies to target exhausted T cells. These magnetic nanoclusters, equipped with PD-1 antibodies, utilize a pH-sensitive bond for attachment and bind to effector T cells through PD-1 receptors. In an acidic environment, they release anti PD-L1 antibodies, blocking PD-1 interactions and maintaining CTL functionality above 90% while delaying tumor progression by over 14 days. The treatment also reduced the abundance of Tregs and increased the abundance of CD8+ CTLs in tumor-bearing mice. When exposed to a magnetic field, these nanoclusters enhanced CTL retention at tumor sites, demonstrating their potential for targeted CTL therapy with minimal impact on physiological parameters, indicating safety [13,40].

Magnetic nanovehicles for targeted drug delivery are designed to enhance the precision and efficacy of cancer treatments. These nanovehicles encapsulate therapeutic agents that are directed to tumors using external magnetic fields, releasing the drug in a controlled manner in response to stimuli such as pH changes, temperature, or enzymatic activity. They also possess theranostic capabilities, acting as contrast agents for magnetic resonance imaging (MRI) to enable real-time monitoring of drug delivery [41,42]. To overcome T cell exhaustion, chaperone cells are used to direct charged nanoparticles to hard-to-reach anatomical compartments [43,44]. The conjugation of liposomes and synthetic nanoparticles with CD8+ T lymphocytes via maleimide-thiol coupling provides continuous pseudo-autocrine stimulation of transferred cells. In models of B16F10 melanoma cultures and metastases in the lung and bone marrow, T cells conjugated with nanoparticles showed 176 times more efficient accumulation in target tissues compared to intravenous nanoparticles, without increasing toxicity or autoimmunity. Additionally, a multilamellar lipid nanoparticle core loaded with IL-15 and IL-21 released cytokines in very low doses over seven days, resulting in significantly higher proliferation compared to systemic infusion. In murine models, all animals treated with nanoparticle-decorated T cells achieved complete tumor clearance and had longer survival than those receiving systemic treatment [16].

#### Antigen Escape and Weakness Transgression

Reduced expression of targeted antigens and the emergence of antigen-negative cell populations account for 9% to 25% of recurrences after CAR T cell therapy [45], due to acquired genetic instability and immunoselection in tumor cells [18]. Despite lymphodepletion pretreatment, residual dying cells are captured by antigen-presenting cells (APCs), which create new MHC Class I and II peptides that prime naive T cells to attack tumor cells [46]. This process, known as epitope spreading, involves antigens recognized by these lymphocytes that differ from those initially targeted by CAR T cells [47]. For effective activation, tumor cryptic antigens must be presented on MHC Class I molecules, engaging cytotoxic CD8+ T cell responses via cross-presentation, primarily performed by specific APCs such as human BDCA3+, XCR1+, and CD141+ dendritic cells [48]. However, the clinical implications of this mechanism and its therapeutic efficacy are not yet fully understood. Some researchers attribute the challenge to inadequate activation of Baft3-dependent dendritic cells, leading to trials of STING agonists in mice to enhance IFN-β responses and improve antigen cross-presentation. In murine models, CAR T cells combined with repetitive STING agonist injections (2′3′-cGAMP) demonstrated strong synergistic effects, achieving a 50% cure rate and reducing contralateral tumor progression. These results largely depended on CD103+ dendritic cells. However, limitations such as intravenous access and uneven drug diffusion and distribution hinder the effectiveness and half-life of STING therapies [46]. Eventually, nanocomplexes could address some of these obstacles. Firstly, using liposomal nanoparticles to deliver cGAMP as STING-L (cGAMP-NP) intracellularly in basal-like triple negative breast cancer and melanoma models resulted in significant tumor growth reduction and increased survival. cGAMP-NP promotes macrophage polarization from M2 to M1, boosts IFN-b and IFN-g levels, and increases the expression of MHC and co-stimulatory molecules [49]. Secondly, artificial APCs can replicate the normal function of natural dendritic cells [50] and be used to enhance antigen spreading after CAR T cell therapy. Yahnhua Li and colleagues have developed hyaluronic acid-modified and carboxylated dendritic mesoporous silica nanoparticles (DMSNs) covalently bonded with functional antibodies (specifically, anti-CD3 and anti-CD28 for activating T cells and anti-PD-1) and tested them in a murine breast cancer model. DMSNs effectively induced T cell activation, as evidenced by increased levels of CD25 and CD69 (T cell activation markers) [51]. The mesoporous silica nanoparticles functionalized with targeting ligands (e.g., antibodies, peptides, and aptamers) and designed to release their cargo in response to stimuli such as an acidic microenvironment or the presence of specific enzymes are known as nanoimpellers. These nanomachines can recognize various antigens on the surface of cancer cells, reducing the likelihood of the cells escaping immune surveillance. Additionally, they can deliver immunomodulatory agents that enhance the immune response against cancer cells, even those that have undergone antigenic changes. When illumination is applied as an external stimulus, the light-activated nanoimpeller increases its dynamic wagging motion, triggering the release of the drug content [52]. Additionally, the dendritic cell-like biomimetic nanoparticles enhanced IFN-γ and TNF-α expression by naive T lymphocytes and exhibited superior therapeutic outcomes in reducing tumor growth [53]. Moreover, the field has explored dendritic cells membrane-coated nanoparticles for cancer immunotherapy applications. Some groups have developed personalized dendritic cell-mimicking nanovaccines (nanoDCs) for stimulation of tumor associated antigens (TAA)-specific T cell populations. NanoDCs were fabricated by coating nanoparticles with membranes from mature bone marrow derived dendritic cells, which were stimulated by tumor cells delivering TAA. NanoDCs successfully reached lymph nodes and primed T cells. The result was a potent tumor size reduction and a decline in metastatic spread [54]. In addition, some groups have designed lipopolyplex platforms to package mRNA molecules into a polymeric polyplex core that is loaded into a phospholipid bilayer shell structure. mRNA has the potential to encode multiple antigens and serves as an adjuvant by triggering Toll-like receptor signaling in the antigen presenting cells. These constructs were efficiently internalized by dendritic cells for antigen presentation and prohibited the mRNA core from interacting with other cells avoiding side effects. Expression of cytokines related to dendritic cell maturation and tumor cell killing was thoroughly promoted [55]. Thus, combination of CAR T cells and these APC-activating nanoparticle technologies may enhance epitope spreading during CAR T cell therapy and reduce post-CAR T cell therapy recurrences due to antigen escape.

### 2.3. Hostile Solid Tumor Microenvironment

The tumor microenvironment significantly influences the efficacy of CAR T cell therapy, often leading to T cell dysfunction and therapy failure. Stromal cells, such as cancer associated fibroblasts activated by tumor growth factor β (TGF-β) produce extracellular matrix proteins that inhibit T cell motility. Furthermore, tumor angiogenesis hampers CAR T cell extravasation into the tumor microenvironment, and disrupts adhesion molecules function, decreasing the effectiveness of the immune synapse. High levels of myeloid derived suppressor cells are related with poor prognosis when CAR T cell therapy for hematological and solid malignancies is used [56,57]. Addressing these challenges, targeting specific pleiotropic cytokines such as TGF-β can improve CAR T cell survival. For example, microenvironment-derived TGF-β inhibits activation, proliferation, and function of cytotoxic T lymphocytes by upregulating of the regulatory gene FoxP3 and by decreasing the production of IFN-γ, perforin, granzymes A and B, and Fas ligand. At the same time, this molecule impairs natural killer function and benefits regulatory T lymphocytes survival [58]. Systemic efforts targeting TGF-β ligands or receptors in clinical trials have shown discrete results and faced several challenges in terms of toxicity and autoimmunity [59]. For this reason, the concept of nano-backpacks is relevant because it utilizes T lymphocytes as vehicles to take loaded nanoparticles to tumor microenvironment. Some authors have explored the possibility of targeting tumor-specific lymphocytes in vivo using liposomes loading a potent small molecule that works as an inhibitor of the TGF-β receptor I (SB525334). In this study, liposomes were targeted to CD90 that exhibits minimal internalization. Through this approach, liposomal TGF-β inhibitor restored granzyme B expression to a higher level compared with the free inhibitor in TGF-β treated cells. Additionally, these liposomes promoted division and expansion of T cells [60].

Li Tang and colleagues investigated a “backpacking” method that chemically links an interleukin 15 superagonist to T cells to enhance their activation in melanoma and glioblastoma mouse models. They developed drug-releasing protein nanogels attached to CD45, which activate upon antigen recognition in the tumor microenvironment. This approach led to a 16-fold increase in T cell expansion within tumors compared to systemic cytokine administration, and a 1000-fold increase compared to T cells without cytokine support. Backpacked T cells proliferated and produced effector cytokines in tumors while remaining inactive in circulation, minimizing toxicity. Overall, this strategy effectively delayed tumor growth [61] (See Figure 2).

Effective interaction between CAR T cells and cancer cells is crucial for adoptive cell therapy. Solid tumors create physical barriers such as dense tissue and compressed vessels, limiting CAR T cell penetration [34,62]. To address this, nanophotosensitizer-engineered CAR T biohybrids (CT-INPs) were developed, using indocyanine green nanoparticles (INPs) attached to CAR T cells. Upon near-infrared (NIR) laser treatment, these biohybrids induced mild photothermal effects without affecting CAR T cell function or viability. Tumor cells were destroyed at temperatures above 43 °C, resulting in 98% tumor cell death. In a mouse model, CT-INPs plus laser treatment reduced tumor growth for up to 40 days. Ultrasound imaging revealed enhanced blood flow and tumor vessel dilation, improving immune cell infiltration and antitumor cytokine expression. The treatment was well tolerated in mice [63].

Drug delivery nanosystems transport treatments to tumors, acquiring a biological identity in vivo (for example, by forming a protein corona). Their size, shape, and surface chemistry influence their movement within the tumor microenvironment, affecting cellular targeting. Passive transport through the EPR (enhanced permeability and retention) effect helps nanoparticles accumulate in tumors, with a size cutoff of 200–1200 nm. However, many are trapped by the extracellular matrix, limiting deep penetration. Smaller nanoparticles (<30 nm) diffuse more easily, while larger ones are often absorbed by tumor macrophages. Nanoparticle size also affects receptor internalization, signaling, and toxicity [64].

The development of nanorobots for cancer treatment emphasizes the biocompatibility of materials to ensure functionality within tumor tissues. DNA origami represents a major breakthrough in nanorobotics [65], while viral capsids, which protect viral nucleic acids and can release them upon binding to specific biomarkers, offer a robust natural design for drug delivery. Materials such as chitosan, gelatin, alginate, pectin, and dextran have been widely applied in cancer therapies for nanoparticle production. A notable innovation is the multi-component magnetic nanorobot, constructed from multi-walled carbon nanotubes (CNTs) loaded with doxorubicin (DOX) and having anticancer antibodies [66]. This nanorobot, driven by an external magnetic field, releases the drug payload in response to intracellular H_2_O_2_ or pH changes, particularly within human colorectal carcinoma cells (HCT116) [9]. Other innovations include pine-pollen-based magnetic microrobots, which deliver DOX inside cancer cells using magnetic rotors, and ultrasound-driven nanowire motors that achieve rapid, near-infrared light-triggered drug release. A tubular, multi-layer nanorobot combines bubble propulsion with magnetic field guidance to efficiently deliver drugs at high speeds, while porous metal rod-like nanorobots carry significantly larger amounts of drugs than their planar counterparts [67]. Janus mesoporous silica nanomotors, cloaked in macrophage cell membranes, allow immune-selective binding to cancer cells, enhancing targeted therapy [68,69]. These advancements in nanotechnology offer potent active drug delivery mechanisms, far surpassing traditional passive systems, and have significant potential for improving cancer treatment.

Another radical approach for treating solid tumors with a hostile microenvironment is to cut off their blood supply through arterial embolization, causing tumor cell necrosis. It has been demonstrated that intravenously injected DNA origami nanorobots (nubots) can induce intravascular thrombosis and inhibit tumor growth in a tumor-bearing mouse model. These customized tubular DNA nanorobots can be bent into a specific conformation. Thrombin is loaded within the tube and isolated from the external environment to prevent enzymatic degradation during transportation. The DNA nanorobot can find its target through specific receptors on the tumor cell membrane surface. Upon reaching the receptor, a built-in molecular switch is activated to release thrombin within the appropriate vessel, blocking the blood vessel and cutting off the nutrient supply to the tumor tissues. However, this study is still in the pre-clinical level [70].

#### Seed and Soil Hypothesis at the Nanoscale

Tumor-derived exosomes, which are extracellular vesicles (40–160 nm) with a lipid bilayer membrane, act as key immune regulators in both the tumor microenvironment and pre-metastatic niches. Their internal or surface bioactive cargo, including proteins, lipids, microRNAs, mRNAs, long non-coding RNAs, and DNA, can disrupt normal immune functions. They decrease the abundance of dendritic cells, inhibit CD8+ T cells, promote the differentiation of immature T cells into regulatory T cells (Tregs), polarize macrophages to an M2 phenotype, suppress NK cell activity, and induce myeloid-derived suppressor cells. These effects help the tumor evade immune detection and prepare distant sites for metastasis [71,72]. Exosomes can also trigger T cell apoptosis through FasL expression [73]. In preclinical models, tumor-derived exosomes have been shown to impair CAR T cell function by increasing inhibitory receptors such as CTLA4 and TIM-3, hindering antigen-specific CAR T cell proliferation and cytotoxicity. This leads to T cell exhaustion, reflected by downregulation of activation proteins and increased expression of exhaustion markers [74]. Exosomes also carry high levels of PD-L1, which further inhibits CAR T cells, reducing their production of granzyme B and IFN-γ. However, blocking exosome activity has been found to enhance CAR T cell effectiveness [75]. On the other hand, CAR T cell-derived exosomes offer therapeutic benefits due to their expression of pro-apoptotic and CAR molecules, giving them cytotoxic properties. Their limited lifespan, lack of replication capacity, and ability to cross tumor barriers, combined with low immunogenicity, make them promising tools for improving adoptive cell therapy by reducing side effects and increasing efficacy [76].

### 2.4. Metabolism and Immune Exclusion

Tumor cells preferentially use aerobic glycolysis for ATP production, converting glucose to lactate despite oxygen availability, which supports their synthesis of key biomolecules and enhances growth [77]. CAR T cells and T lymphocytes also rely on this metabolic pathway, leading to competition for glucose in the tumor microenvironment [78,79]. Aerobic glycolysis is essential for T cell effector function, particularly for translating IFN-γ mRNA. However, the glucose-restrictive environment caused by rapidly proliferating tumor cells can metabolically block T cells, reducing IFN-γ production and leading to decreased proinflammatory cytokines and T cell hyporesponsiveness over time [80]. Other molecules, such as arginine, tryptophan, and reactive oxygen species, further disrupt T cell metabolism and contribute to immune evasion [81,82]. Activated T cells differentiate into various subsets: effector T cells (Teff), stem cell memory T cells (Tscm), central memory T cells (Tcm), effector memory T cells (Tem), and tissue resident memory T cells (Trm), each with distinct capabilities [83,84]. Less differentiated phenotypes, such as stem cell memory and central memory T cells, which have greater potential for self-renewal and resistance to exhaustion, are associated with better clinical responses in CAR T cell therapies [36,85].

Researchers are striving to balance the killing capacity, expansion, and persistence of CAR T cells. Cytokines such as IL-7, IL-15, and IL-21 promote oxidative phosphorylation, aiding the expansion of Tscm-like and Tcm-like cells, which are linked to long-lasting anti-tumor activity due to their stemness and persistence [86,87]. Nano-delivery systems have emerged to efficiently transport cytokines or regulate their expression in tumor cells, lowering the required dose and reducing adverse effects while protecting the cytokines from degrading before reaching their target [88]. For instance, poly-γ-glutamic acid-based platforms with chitosan have enhanced cytokine secretion by macrophages, increasing IL-6, IL-12, and TNF-α levels, which inhibit tumor cell invasion [89]. Additionally, β-cyclodextrin-based nano-systems and adenovirus vectors carrying the IL-12 gene have been shown to inhibit tumor growth in mouse melanoma models. Liposome-based nanomaterials, valued for their histocompatibility and modifiability, have also been used to deliver cytokines, for example, by delivering mRNAs encoding IL-12 and IL-27, to modulate the tumor microenvironment (TME), boosting IFN-γ and TNF-α levels and activating NK cells and cytotoxic T lymphocytes (CTLs) [90]. Similarly, inorganic nanomaterials, including mesoporous silica nanoparticles, magnetic nanoparticles, and gold-based structures, have been explored as potential delivery systems [91]. Additionally, anti-PD-L1 therapy has been found to reduce tumor cell expression of glycolysis enzymes and mTOR protein phosphorylation, increasing glucose availability for T cells, independent of PD-1 expression [92]. The binding of PD-L1 to PD-1 inhibits glycolytic activity and mitochondrial biogenesis, reducing glucose for infiltrating T cells [92,93]. Nanomachines further enhance local concentrations of anti-PD-L1 molecules, supporting the use of CAR T cell nanosymbionts to regulate the metabolic environment in adoptive cell therapy.

Alternative ways of failure of CAR T cell therapy and resistance can also be explained by certain metabolic conditions. Trogocytosis is a cell phenomenon where one acceptor cell acquires and internalizes the membrane of donor cells including their surface molecules [94]. Acceptor cells can adopt behaviors such as those of the donor cells [95]. CAR molecules also suffer this destiny, leading to CAR T cell death and antigen density reduction [96,97]. You Zhai and colleagues demonstrated how CAR T cells become donors with loss of their receptors, meanwhile tumor cells acquire CAR molecules as acceptors via the IS. Resistance occurs because antigen-dependent cytotoxicity of these CAR T cells becomes weaker and tumor cell antigen density is reduced. This group also found that trogocytosis is associated with cholesterol metabolism in tumor cells. Statins were used in tumor cells to inhibit CAR molecule transfer with successful preservation of cytotoxicity [98]. Nanocarriers loaded with simvastatin including superparamagnetic iron oxide nanoparticles, poly (D,L-lactide-co-glycolide) (PLGA) with cholic acid nucleus nanoparticles, alpha-lipoic acid (ALA) nanoparticles, liposomes, and nanoemulsions have shown better bioavailability, cellular uptake, cytotoxicity, and selectivity than systemic use [99].

### 2.5. The Hurdles of Living Drug Tracking

CAR T cell therapy is a new biotechnology with different biological checkpoints involved in its mechanism of action, different toxicity profile, overall responses, and patterns of failure and resistance. Strategies currently used by clinicians to follow activity of infused CAR T cells consist of flow cytometry, immunohistochemistry, and quantitative polymerase chain reaction (qPCR) for peripheral blood, bone marrow, tumors, and lymph node biopsies are not practical in most clinical trials. They are invasive and do not provide real-time whole-body spatio-temporal distribution of infused T cells without a comprehensive picture of CAR T cell activity [100,101,102].

There is a clinical need for a technique that can monitor in vivo performance of CAR T cells in tumors and off-target sites. Louise Kiru et al. developed a MRI-based cell tracking technique with potential for clinical translation using ferumoxytol (iron oxide nanoparticles detected by MRI) as a cell marker to monitor real time in vivo CAR T cell trafficking in preclinical osteosarcoma model. Ferumoxytol was used with a microfluidic device for mechanoporation-labeling of human anti-B7-H3 (a tumor antigen up regulated in osteosarcomas) CAR T cells. These CAR T cells were identified through MRI, photoacoustic tomography (PAT), and magnetic particle imaging (MPI). Mechanoporation did not alter the proliferation or function of T cells. At week 1, the tumor demonstrated iron enhancement on T2-weighted MRI only in the ferumoxytol-labeled B7-H3 CAR T cell group indicating enhanced infiltration of the T cells in the tumor tissue. This group found a significant correlation between tumor T2 values at week 1 and decrease in tumor size at week 3. MPI and PAT images after infusion of B7-H3 CAR T cells confirm infiltration of CAR T cells in tumor tissue [103]. Others have performed similar studies using different magnetic resonance contrast agents (e.g., ferucarbotran) for MPI-labeled T cell detection in the brain of tumor-bearing mice sparing T cell viability [104].

Approaches to radiolabel-modified T cells have also been tested. Using the Sleeping Beauty transposon/transposase system, Bhatnagar and colleagues designed CD19-specific CAR in a murine model. T cells that co-express CD19 CAR and luciferase were loaded with gold nanoparticles (GNP) (particles appropriate for intracellular retention) functionalized with copper-64 (^64^Cu^2+^) as a PET reporter and polyethylene glycol (GNP-64Cu/PEG2000). Copper-64 (^64^Cu^2+^) was chosen as radioisotope for PET, based on its longer half-life. These CAR T cells were infused into mice, and PET imaging in the transverse, coronal, and sagittal planes was performed. Only mice with CAR T cells bearing GNP-64Cu/PEG2000 showed longer-term PET signals. These PET signals colocalized with bioluminescent imaging (BLI) signal, confirming the possibility of tracking CAR T cells using positron emitter imaged by PET/CT scanner [105]. Others have also tried non-genomic ex-vivo labeling of CAR T cells with radionuclides. Harmsen et al. employed a dual-modal PET/near infrared fluorescence (NIRF) silica nanoparticle as non-genomic labels for human carcinoembryonic antigen (hCEA)-redirected CAR T cells in tumor-bearing mice. Zirconium (89Zr) was chosen as a radioisotope for PET. Mice were injected intraperitoneally or intravenously. PET/NIRF nanotag-labeled CAR T cells retained their functional activity. After two weeks of the intraperitoneal injection, the BLI signal of the CAR T cells was mainly in a single focus, and there was a discrepancy with the PET image. This indicated that PET/NIRF nanotags were released from CAR T cells at week 1 and accumulated in tumor cells. This approach showed whole-body tracking of CAR T cells for up to 1 week and an alternative way to improve tumor accumulation of nanoparticles (see Figure 2) [106].

Thus, these technologies can reveal the in vivo dynamics of both CAR T cells and nanoparticles, promoting further understanding of CAR T cell biology in vivo and optimal design of CAR T cells and nanoparticles.

## 3. Engineering CAR T Cells: Current Design Roadblocks in Perspective

The production of CAR T cells involves several crucial steps. The major aspects of the CAR T cell manufacturing process are relatively standardized, whereas clear differences can be identified in every single manufacturing step [42]. Firstly, T cells are collected from the patient through leukapheresis, enriched, and activated using specific stimuli to enhance their proliferation and responsiveness. Subsequently, a gene encoding the CAR is introduced into these activated T cells using either viral vectors or non-viral methods. These modified T cells are then cultured under optimal conditions to promote their expansion and multiplication, after which those T cells are selected and purified to ensure the elimination of undesired cells. In the final step, cryopreservation of the CAR T cells is carried out [37]. Quality control testing is performed during the production as well as for the final cryopreserved CAR T cell product for the integrity of the product [107]. It is important to note that slight variations in these steps may exist depending on the specific manufacturing facility and the technologies employed [37] The average duration of GMP CAR T cell manufacture is 12 days (range, 7–22 days) [41] (see Figure 3). We will now explore different issues and challenges that arise in the manufacturing of CAR T cells, diving into each individual step and then analyzing the problems in the current and broader context.

The initial step in CAR T cell therapy involves collecting peripheral blood mononuclear cells (PBMCs) via leukapheresis. A significant challenge here is the variability in the starting material due to patient-specific factors such as age, disease stage, and prior treatments, which can compromise T cell quality and function. For instance, patients with non-Hodgkin lymphoma or acute lymphoblastic leukemia often have decreased concentrations of memory T cells after standard treatments, leading to manufacturing failures or suboptimal CAR T cell function [108,109]. Cryopreservation further reduces the viability and recovery of PBMCs compared to fresh products. Additionally, logistical limitations with fresh apheresis products, such as narrow viability windows for manufacturing, complicate the process. Variability introduced by different apheresis reagents, instruments, and personnel can also affect the quality of the collected cells. These challenges highlight a need for technologies that can standardize and preserve T cell quality during collection and storage [110].

Post-leukapheresis, contaminants such as platelets, plasma, and residual anticoagulants can adversely affect T cell activation and expansion. Current separation methods, such as immunomagnetic separation using antibody-coated beads, are costly and complex and may introduce additional contaminants or inadvertently activate T cells prematurely, leading to rapid exhaustion or tonic signaling [108,111,112,113]. There is also a risk of unwanted cell types, such as NK cells, contaminating the product. These methods require GMP-grade reagents and rigorous protocols to prevent undesirable effects, adding to the complexity and cost. The need for label-free, efficient, and less cumbersome T cell enrichment methods presents an opportunity for nanotechnology-based solutions that can selectively isolate T cell subsets without the drawbacks of current techniques [114].

Effective T cell activation is crucial for successful CAR T cell manufacturing [115]. Traditional methods involve using anti-CD3/CD28 antibody-coated paramagnetic beads, which must be meticulously removed before infusion to avoid adverse patient reactions. This bead removal process is labor-intensive, time-consuming, and increases the risk of contamination, requiring multiple operators and careful handling [108,116,117,118,119,120]. Alternative activation methods, such as soluble activation reagents or artificial antigen-presenting cells (APCs), exist but may still have limitations regarding efficiency, scalability, or cost [121]. The variability in activation protocols across different centers leads to inconsistencies in the final CAR T cell products, affecting clinical outcomes. There is a clear need for innovative activation strategies that are efficient, scalable, and reduce variability—an area where nanotechnology could offer significant advancements [121,122].

Gene delivery is a critical step, traditionally achieved using viral vectors such as retroviruses and lentiviruses [121]. While effective, these methods pose safety risks such as insertional mutagenesis and potential oncogenesis due to random integration into the host genome. Viral vector production is also expensive; labor-intensive; and subject to batch variability, leading to inconsistent transduction efficiencies (ranging from 4% to 70%). Non-viral methods, such as electroporation of plasmid DNA or transposon/transposase systems (e.g., Sleeping Beauty), offer alternatives but have limitations, including lower efficiency, extended culture times, and potential genomic instability [123,124,125,126]. Moreover, CRISPR-Cas systems for gene editing present challenges in delivery efficiency and safety concerns such as off-target effects. These issues highlight the necessity for safer, more efficient, and cost-effective gene delivery methods [108].

Researchers are employing gene-carrier nanoparticles to efficiently express chimeric antigen receptors (CARs) in T cells while minimizing toxicity to target cells [127,128]. This innovative approach includes reprogramming T cells directly within the body using nanocarriers, which eliminates the need for external cell manufacturing processes. Studies have demonstrated that CD3-targeted nanoparticles carrying plasmid DNA can deliver leukemia-specific CAR genes to T cells in vivo, resulting in disease remission. Further research using CD3- and CD8-targeted nanocarriers loaded with in vitro-transcribed mRNA has achieved effective CAR expression in T cells, leading to disease regression in mouse models of leukemia, prostate cancer, and hepatitis B-induced liver cancer [129].

Another study focused on creating antifibrotic CAR T cells that target fibroblast activation protein (FAP), a marker of fibroblast activation. Delivering FAP-CAR-mRNA via CD5-targeted lipid nanoparticles reduced fibrosis and improved heart function in rodent models with tissue scarring. These nanoparticle-based systems offer enhanced manageability and precise timing in clinical applications, positioning them as promising off-the-shelf solutions for various medical conditions [96]. Despite these advancements, concerns about long-term genomic safety persist, including issues such as genomic insertion and promoter dependency. While nanoparticles provide comparable treatment results to traditional viral vectors and offer benefits such as simplified storage and reduced costs, these safety issues need thorough investigation [130]. Cationic polymers such as polyethyleneimine (PEI) and poly (2-dimethylaminoethyl methacrylate) (pDMAEMA) are used to form complexes with nucleic acids, facilitating their entry into cells by crossing the cell membrane [131]. Nanoparticles typically deliver their genetic cargo via endocytosis or membrane fusion, attaching to sulfated proteoglycans on T cell membranes. Nanoparticle-sensitized photoporation is another efficient method that opens pores on cell surfaces to allow external cargo entry, capable of producing engineered T cells at a high throughput exceeding 10^5^ cells per second.

Lipid nanoparticles are commonly used for nucleic acid delivery because they can encapsulate large nucleic acid molecules and protect them from degradation. Lipidoids, a subtype of lipid nanoparticles, are easily prepared and share many properties with lipids. Combining nanomaterials with electroporation techniques may enhance the transfection efficiency of large DNA plasmids into human primary T cells. A novel, stimulus-sensitive cationic nanomicelle based on a specific block co-polymer has shown efficiency in delivering and releasing DNA to targeted sites [132]. Overall, while nanoparticle-based methods show significant promise for producing CAR T cells and treating various diseases, further research is necessary to fully assess their safety and effectiveness, especially for in vivo applications [133,134].

Scaling up CAR T cell production requires expanding transduced cells to clinically relevant numbers. Traditional static cultures are impractical due to labor intensity and high contamination risk [108,121]. Bioreactors such as the G-Rex system and rocking-platform bioreactors (e.g., GE WAVE) offer improvements but come with their own challenges [135,136,137]. The G-Rex system can disturb cell cultures during sampling, affecting expansion kinetics, while rocking-platform bioreactors are susceptible to mechanical failures and contamination risks due to semi-automated operations [135,136,138,139,140]. Fully automated systems such as CliniMACS Prodigy aim to streamline the process but are limited by their inability to process multiple batches simultaneously, potentially slowing down production [135]. These limitations point to a need for advanced, fully automated, and scalable expansion technologies that ensure consistent cell quality and reduce contamination risk. Nanotechnology could contribute to developing microfluidic bioreactors or nanoscale scaffolds that provide a controlled environment for efficient T cell expansion [141].

Before infusion, CAR T cell products undergo cryopreservation and extensive quality control testing to meet GMP standards. However, studies show that a notable percentage of manufactured products fail to meet release criteria due to variability in cell numbers and quality. The GMP process also imposes a heavy documentation burden and navigational challenges within a varied regulatory landscape. These complexities can delay treatment and increase costs. There is a pressing need for technologies that simplify quality control processes and enhance the consistency and stability of the final CAR T cell product. Nanotechnology may offer advanced biosensing and analytical tools for more efficient quality assessments [110].

### 3.1. The Issue of Viral Vectors

Vectors are the usual method for gene transfer in CAR T cell production due to their efficiency in delivering genetic material to target cells [142]. They are classified into viral (VV) and non-viral (NVV) types, with lentiviruses (LVs), derived from HIV-1, being the most common in CAR T therapy. LVs integrate genetic material into the host cell by transducing the gene of interest [143]. To produce LV vectors, HEK293T cells are transfected with plasmid DNA (pDNA), leading to the assembly and release of viral particles into the culture medium [144]. This process is cost-effective compared to adeno-associated viruses as it does not require a lysis step [145]. LVs are genetically modified to increase their genetic payload, reduce pathogenicity, and prevent replication [58], though they face limitations in safety, cost, and flexibility compared to non-viral nano transfer systems.

#### 3.1.1. Safety Concerns

Lentivirus (LV) vectors are designed with separate plasmids to prevent competent viral replication. However, integrating transgenes into the genome can raise the risk of insertional mutagenesis, which varies by vector type [146]. Despite this, genotoxicity remains rare, and LV vectors are considered safer than γ-retroviruses due to their tendency to integrate into transcribed genes [147]. There are no reported cases of oncogenic transformation linked to LV T cell transduction, but three instances of insertional mutagenesis exist in the literature. Shah et al. reported clonal expansion in a patient treated with anti-CD22 CAR T cell therapy, resulting in remission [148]. Fraietta et al. found CAR T cell expansion in a patient receiving anti-CD19 CAR T cell therapy, linked to LV integration in the TET2 gene [149]. Lastly, Cavazzana-Calvo et al. described clonal expansion in a patient with thalassemia, caused by LV integration into the HMGA2 gene, but the clone eventually dissipated [150]. Some alternatives to prevent genotoxic issues include using transgenes with “suicide genes” or OFF-switches to control CAR T cell activity without harming unmodified cells [151]. One example is the herpes simplex virus-thymidine kinase (HSV-TK) system [152], which requires prodrugs such as ganciclovir for activation but poses challenges such as slow activation and immunogenicity [153,154]. Another approach is using mRNA for CAR expression, which reduces side effects [155] but requires repeated treatments due to its short-lived expression [156].

#### 3.1.2. Transduction Efficiencies

Quantification of transfection capability is another challenge to cover when using VV, mainly because there is lot-to-lot variability that contributes to inconsistency in therapeutic T cell production; the expression rate ranges from 5 to 39% of CAR T cells, causing the number of delivered nucleated cells to range from 1.7 × 10^8^ to 50 × 10^8^, which causes the subsequent therapeutic efficacy to vary as well [157]. On the other hand, non-viral vectors yielded more uniform and consistent CAR expression in >40% of the T cells, showing superior anti-tumor activity [158].

#### 3.1.3. Accessibility

Viral production for clinical applications under Good Manufacturing Practice (GMP) standards takes 2–3 weeks and requires highly regulated processes to ensure product safety and quality. GMP involves the use of biosafety level 3 (BSL3) clean rooms, stringent safety testing, and trained staff to maintain controlled and reproducible conditions [110]. However, a limited number of third-party suppliers monopolize viral vector production, raising concerns about access and equity. FDA guidelines mandate extensive testing to prevent the occurrence of replication-competent viruses during vector production and ex vivo cell therapy product release, with long-term follow-up of up to 15 years [159]. These regulatory requirements increase costs and reduce global availability. In contrast, non-viral gene transfer offers a more efficient alternative, with faster production, lower costs, and no risk of viral replication [59]. CAR T cell therapies, for instance, rely on costly lentiviral vectors, which require custom packaging and strict temperature control and can cost between $950 and $1250 USD for RNA lentiviral vectors [74]. Non-viral systems could replace these with cheaper and faster methods, improving accessibility.

### 3.2. CAR T Nanosymbionts Overcoming Toxicity of CAR T Cell Therapy

Current standard design of CAR T cells can generate unpredictable outcomes regarding toxicity because of its own capacity to expand, persist, amplify its response, and engage with several cells and tumor-associated antigens in non-malignant and malignant tissues. Cytokine release syndrome (CRS), immune effector cell-associated neurotoxicity syndrome (ICANS), and on-target, off-tumor toxicity (OTOT) represent the most common toxicities related to CAR T cell therapy. Between 42 and 100% and up to 46% of patients, respectively, develop any grade of CRS and severe CRS after CAR T cell infusion [160,161,162]. CRS is explained by overactivation of effector cells with subsequent production of massive levels of proinflammatory cytokines, and endothelial cell activation and damage [163,164,165]. Injured endothelial cells increase vascular permeability leading to edema, organ hypoperfusion, coagulopathy, and organ dysfunction [166]. Regarding ICANS, its incidence varies from 2% to 64% in general and measures up to 50% for severe symptoms [77,78]. There is activation of the cerebral vascular endothelium and impairment of the brain blood barrier (BBB). Cytokines alter astrocytes and pericytes, leading to cerebral edema, thrombosis, hemolysis, and even disseminated intravascular coagulation [167,168]. There are also interesting data showing on-target off-tumor effect as a part of the pathophysiological mechanism of ICANS. CD19 is expressed on pericytes and vascular smooth muscle cells and can be recognized by CAR T cells with subsequent BBB dysfunction [169].

The amount of CAR T cells infused to patients is directly related with efficacy and toxicity. Low-dose infusions can decrease CRS and ICANS development but increase the risk of relapse and failure of treatment [170]. Some authors have proposed the concept of fractionated dosing, delivering T cells several times with dose scalation and the possibility to stop infusions when toxicity appears. Those studies concluded that the fractionated administration can be considered safe and produced efficacy results comparable with other academic or even commercial products. Despite that, these analyses are limited by the small sample sizes and trial design and require further explorations in phase 2 trials [171,172]. A rational alternative approach is packaging modified mRNA in nanoparticles to generate CAR T cells through endocytosis by using endosomal escape [173]. Targeting antibodies on nanoparticles surface are used to guide mRNA transcription inside specific cell types. This method brings several advantages. There is no genomic integration, for that reason, CAR T cells produced in this way are transient, providing the ability to titrate dosing and to re-dose as needed. Appropriate regulation and sequence optimization can offer a significant increase in mRNA stability and translation efficiency. It only requires short development period. A new mRNA nanoparticle could be rapidly redesigned and produced based on a new genetic material sequence. Finally, there is a wide range of applications for this technology (cancer, infections, autoimmune, cardiovascular, and metabolic diseases [173]. This approach, for example, has been used to develop transient antifibrotic CAR T cells in vivo (already shown above) demonstrating efficacy to deliver modified mRNA encoding the CAR with subsequent cardiac fibrosis reduction [88] (see Figure 2). Perhaps CAR T nanosymbionts using mRNA can also help to alleviate recent fears concerning the safety signal announced by the FDA Center for Biologics Evaluation and Research related to T cell lymphoma development in several patients undergoing CAR T cell therapy for pediatric acute lymphocytic leukemia, non-Hodgkin lymphoma, and multiple myeloma with malignant clones containing the genetic signature of the CAR construct [147,174].

Certain characteristics of nanoparticle surfaces can reduce overall accumulation in off-target organs [175]. Because of the EPR effect alone, the use of nanoparticles results in almost a 10-fold increase in molecule retention at the tumor site [176,177]. Given that nanoparticles preferentially enter the extravascular space of the tumor and remain within the tumor site [177], this capacity can be used to carry mRNA to induce CARs in tumor infiltrating lymphocytes to decrease OTOT. In addition, nanotechnology can help to optimize CAR and TCR (T cell receptor) functionality through the use of in vitro transcribed (IVT) mRNA encoding TCRs or CARs with different specificities on the same T lymphocyte increasing target recognition and decreasing normal tissue damages [178,179]. In general, OTOT is not a problem for hematological malignancies, and the consequences, such as B cell aplasia and hypogammaglobulinemia in the context of CD19-directed CAR Ts, are manageable [180]. Conversely, in solid tumors, target antigens (also known as tumor-associated antigen: TAA) are often co-expressed in normal tissues. After antigen recognition, there are triggered CAR T effector functions with perforin and granzyme release, upregulation of Fas ligand, and secretion of IFNγ and TNF, causing normal tissue destruction [181,182]. Several TAAs have been targeted in solid tumor CAR T trials. CAIX, CEACAM 5, HER2, EGFR, and CLDN18.2 have been tested with different grades of liver, lung, dermal, and gastrointestinal toxicity [183,184,185,186]. To overcome this challenge, several strategies have been implemented. One of them disengages CAR T cell activation signals using “dual AND-logic” CARs with two distinct synthetic receptors expression. Both antigens are needed to produce CAR T cell activation and, as a result, an increase in specificity [146]. Although useful, this perspective does not resolve the risk of tumor immune escape due to antigen loss or negative antigen selection [187]. A possible contribution of nanotechnology would be to help CAR T cells recognize specific patterns derived from tumors. Sukumaran and colleagues modified the “dual AND-logic” concept reshaping CAR T cells to be responsive to nominally immunosuppressive soluble factors produced by both the tumor and stroma that contribute to tumor growth using a trio of receptors for prostate stem cell antigen (PSCA), TGF-b, and IL4, leading to “Smar T cell” activation and long-term persistence and avoiding antigen escape [188]. Controlled release of soluble cytokines can be achieved with biodegradable nanoparticles using materials such as PLGA [189] as well as hydrogels and liposomes [190]. Furthermore, these Smar T cells may also be conjugated with nanoparticle backpacks providing local cytokine subsistence to strengthen the third immunological signal directly related to cell persistence [191].

Nanotechnology can also contribute to turning T cells off at non-tumor sites. There is evidence of reversible control of CAR T cells with the use of tyrosine kinase inhibitor dasatinib (DAS), which prevents phosphorylation of immunoreceptor tyrosine kinase-based activations motifs (ITAMS) within CD3ζ, interrupting one of the two signals for CAR T cell effector function leading to control of acute CAR T toxicities [192]. The key issues with this scheme are that systemically given DAS has poor water solubility, its absorption is easily affected by the pH, and its terminal half-life is only 3–4 h [193]. Nano drug delivery systems offer a targeted release of DAS and increase its intracellular accumulation to provide safe and efficient DAS delivery. Zhang from the School of Pharmacy at Bengbu Medical College, China, designed a pH-sensitive targeted nanoparticle linking the hydroxyl group of DAS and the carboxyl group of hyaluronic acid (HA). HA acts as a ligand for CD44 and in this way can recognize target cells. The DAS-HA complex is broken in response to acidic microenvironment in tumor stroma.

## 4. All Are Not Hunters That Blow the Horn

The following issues are some of the obstacles that must be resolved before nanotechnology can boost CAR T cell therapy.

### 4.1. Technical Characteristics of Nanosystems

#### 4.1.1. Nanomaterial Sterility

Sterility is mandatory to avoid immunostimulatory reactions in nanoformulations [194]. Endotoxin contamination assays with the capacity to detect physical presence and biological activity are needed, but confirmatory tests must be performed to avoid high false positive rates. Purification strategies can remove residual manufacturing components to avoid toxicity and improve the efficacy of the mRNA being delivered [195].

#### 4.1.2. Biocompatibility of Non-Active Pharmaceutical Ingredients

Nanomaterials should either degrade into non-toxic byproducts that can be easily eliminated by the body or remain inert without interfering with normal biological functions. Biodegradable materials are particularly preferred as they reduce long-term accumulation and potential chronic toxicity [76].

#### 4.1.3. Impact of Nanoparticles on the Immune Response

Some nanoparticles have shown inhibition of natural killer cell activity, monocytes, and macrophages with suppression of proinflammatory cytokine production [188]. Additionally, liposomes can trigger an innate immune response, leading to an acute hypersensitivity syndrome known as complement activation-related pseudoallergy (CARPA) [190]. To mitigate these issues, surface modification and charge optimization of nanoparticles with appropriate agents are necessary. For example, coating nanoparticles with polyethylene glycol (PEG) helps create a “stealth” layer that reduces recognition and uptake by immune cells. PEGylation can decrease protein adsorption on the nanoparticle surface, which, in turn, reduces opsonization and subsequent clearance by macrophages [196]. Adjusting the surface charge of nanoparticles can also influence their interaction with immune cells [197].

#### 4.1.4. Clearance and Agglomeration

Nanoparticles administered systemically often face inefficient distribution due to rapid clearance by the mononuclear phagocytic system (MPS), renal filtration, opsonization, and tissue agglomeration [198,199,200]. To enhance their concentration at target sites such as tumors, solutions involve modifying nanoparticle design. Adjusting size and shape can reduce rapid kidney clearance; particles larger than 6 nm are less likely to be eliminated [201]. Neutralizing surface charge and applying surface modifications such as polyethylene glycol (PEGylation) can minimize clearance by the liver, spleen, and immune system [202]. Using biodegradable materials ensures nanoparticles degrade safely, preventing long-term accumulation and associated safety risks such as nephrogenic systemic fibrosis and central nervous system diseases observed with persistent nanosystems (e.g., certain quantum dots) [203].

#### 4.1.5. Drug Release

Some nanosystems used as backpacks have important limitations such as low drug loading capacity and a lack of modulation of drug release [204]. Researchers have developed targeted drug delivery systems that enhance therapeutic efficacy while minimizing off-target effects. This approach involves incorporating targeting ligands—such as antibodies, peptides, and aptamers—onto nanoparticle surfaces, which specifically recognize and bind to tumor cell receptors. By enabling nanoparticles to selectively target tumor cells, these systems improve drug delivery efficiency and reduce side effects [205].

#### 4.1.6. Tumor Microenvironment and Nanoparticles Effect

Nanoparticle parameters may vary with microenvironment conditions. Changes in size and charge can modify biodistribution, toxicity, and immunological circumstances [206,207,208,209].

#### 4.1.7. Batch Consistency in Clinical Use

Different aspects of nanoparticle must be considered to validate its quality in the clinic. Size, aggregation, charge, stiffness, biological activity, cargo release rates, and other factors will have certain levels of heterogeneity that must be defined to establish adequate ranges to release and, at the same time, to reject them [210]. Finally, transition from benchtop to bedside for a considerable number of nanosystems is still a bottleneck mainly because of insufficient knowledge about functional aspects of nanomaterial composition, size, shape, elasticity, and surface charge when in contact with biological components [97].

#### 4.1.8. Surface Decoration of Some Nanoparticles

PEGylation enhances liposome pharmacokinetics by preventing cell interactions and promoting tumor accumulation via the EPR effect [197,198]. However, it can cause accelerated blood clearance (ABC) due to anti-PEG antibodies, impairing nanoparticle circulation [199]. To address this, strategies include modifying administration regimens, using alternative or cleavable PEGs, reducing PEG chain density and length, and attaching targeting ligands to improve specificity and efficacy [200,201,202,211,212]. Other approaches involve applying zwitterionic coatings such as phosphorylcholine or sulfobetaine, incorporating biodegradable polymers such as PLGA, chitosan, or PCL, using non-ionic surfactants such as Tween 80 or Pluronic F68, and adding hydrophilic polymers such as dextran or hyaluronic acid to enhance stability and circulation time [213,214,215].These methods collectively aim to improve the safety and effectiveness of nanoparticle-based drug delivery systems.

### 4.2. Pricing/Challenges for Low- and Middle-Income Countries

The production of CAR T cells for conventional medical therapies involves significant costs and presents distinct challenges in terms of processing and access. For CAR T cells, the manufacturing process is highly specialized and expensive, requiring specialized facilities and highly trained personnel. This contributes to the high costs associated with CAR T therapies, with a median cost of $475,000 USD per patient [216]. When considering the entire process, including production, transfusion, biopsies, hospital stays, and imaging studies, the total cost can rise to between $500,000 and $1,000,000 USD [203]. Nanomedicine, on the other hand, involves using nanoparticles to deliver drugs or other therapeutic agents to specific targets in the body. Nanoparticles can be manufactured in large quantities using relatively simple methods, making them more cost-effective to produce than cell therapies [99]. The cost of nanomedicines for clinical use varies depending on the specific agent and application but is generally lower than the cost of CAR T cell therapy, with some treatments priced around $80,000 USD [121]. It is expected that nanotechnology will play a significant role in CAR T immunotherapy for cancer, potentially reducing the cost of this treatment. Nanotechnology might facilitate the use of non-viral vectors for gene delivery, such as lipid nanoparticles (LNPs) or polymeric nanoparticles, which are generally cheaper to produce and handle than viral vectors and can simplify the manufacturing process. Additionally, nanoparticles can be used to deliver CAR constructs directly to T cells in the patient’s body, potentially eliminating the need for ex vivo cell manipulation. This approach can significantly reduce manufacturing complexity and costs. In low- and middle-income countries, the high cost of CAR T cell treatment is a growing issue, restricting its use in most emerging economies’ healthcare systems. Nanoparticles could help reduce CAR T cell costs, making the process more affordable without compromising efficacy [36].

Producing CAR T cells for medical therapies is expensive and complex, requiring specialized facilities and highly trained personnel, which drives the median cost up to $475,000 per patient and can total between $500,000 and $1 million when including additional medical expenses [98,203]. Nanomedicine offers a more cost-effective alternative, as nanoparticles can be manufactured in large quantities using simpler methods, with some treatments costing around $80,000. Nanotechnology is expected to play a significant role in CAR T cell immunotherapy by reducing costs—using non-viral vectors such as lipid nanoparticles or polymeric nanoparticles for gene delivery can simplify the manufacturing process and are cheaper than viral vectors. Additionally, nanoparticles can deliver CAR constructs directly to T cells within the patient’s body, potentially eliminating the need for ex vivo cell manipulation and further reducing costs. This approach could make CAR T cell therapies more accessible, especially in low- and middle-income countries where high costs currently limit their use.

### 4.3. Scale-Up

Although the global nanomedicine market is expected to grow from $246.2 billion in 2021 to $493.5 billion by 2026, few nanosystems reach clinical use due to challenges in scaling-up production [217]. Precisely controlling the size, shape, and composition of nanomolecules becomes difficult on a larger scale, leading to variations that can affect performance and increase production costs [218]. Advanced software for process optimization and specific analytical methods—such as dynamic light scattering, zeta potential measurement, and various microscopy techniques—are essential for maintaining quality and precision during scale-up [219,220]. Additionally, sterilization methods can alter nanoparticle characteristics, and environmental safety concerns arise from potential pulmonary toxicity due to nanoparticle exposure [221]. Efforts must focus on minimizing environmental release, ensuring proper disposal, and conducting thorough toxicity testing to predict immune interactions [222,223,224]. The regulatory landscape is underdeveloped, highlighting the need for international standards, critical quality attributes, and specific non-clinical tests [225,226,227]. Improving risk–benefit assessments, traceability of nanomaterial-containing health products, and pharmacovigilance systems is imperative for the responsible advancement of nanomedicine [228].

## 5. Materials and Methods

In this literature review, a comprehensive search and analysis of existing articles, books, and relevant publications were conducted to synthesize current knowledge on the topic. The review process began with the identification of key themes and concepts, followed by the selection of databases such as MEDLINE, Embase, Google Scholar, and Cochrane for source retrieval. Keywords and Boolean operators were strategically employed to refine search results, ensuring the inclusion of studies published within the last ten years to maintain relevance. Each selected piece of literature was critically evaluated for methodological rigor, theoretical contributions, and relevance to the research questions. The findings were then organized thematically, allowing the identification of trends, gaps, and areas of consensus in the existing body of work. This methodical approach ensured a thorough and unbiased examination of the literature, providing a solid foundation for the subsequent analysis.

## 6. Conclusions

In essence, CAR T nanosymbionts are currently conceptual frameworks under development. Combining the two technologies holds promise for optimizing their respective outcomes. There is a critical need to streamline the development of adoptive cell therapy to broaden its accessibility and overcome current hurdles. Potential strategies include replacing the ex vivo CAR T cell manufacturing process with an in vivo approach, utilizing nanotechnology-based non-viral vectors, employing mRNA instead of DNA, and incorporating adjuvant nanomolecules after CAR T infusion. These approaches, among others in research, aim to achieve the aforementioned objectives. Various nanosystems can exploit intrinsic CAR characteristics. Looking forward, advanced medical nanorobots are poised to perform diverse medical functions, particularly in modulating the microenvironment of solid tumors, thereby enhancing CAR T immunotherapy. These nanorobots are envisioned to evolve into sophisticated “nanosubmarines” in the bloodstream, customizable and programmable based on individual patient and tumor biology. Nanoparticles are an ideal platform for combination therapies, where cancer vaccines can be used alongside other immunotherapies, such as checkpoint inhibitors or CAR T cell therapies. Nanoparticles can help co-deliver these agents in a synergistic manner, potentially leading to a more robust and comprehensive anticancer response. This customization allows personalized treatment strategies that maximize therapeutic efficacy while minimizing side effects. Strategies such as refining apoptosis through bystander cytotoxicity, enhancing immune signaling strength via differential co-stimulatory domain expression to boost potency, and addressing resistance through enhanced IFNγR-related cell adhesion pathways are being explored. Simultaneously, targeting exhausted T cells using nanoclusters equipped with immune checkpoint inhibitors or pro-inflammatory cytokines, promoting epitope spreading, and employing artificial antigen-presenting cells or cell membrane-mimicking nanoparticles for T cell priming are emerging concepts with potential clinical applications. Addressing adoptive cell therapy toxicity involves reshaping CAR T cells to recognize tumor-derived patterns and employing biodegradable nanoparticles for controlled release of soluble cytokines, minimizing antigen escape risks. Additionally, targeted delivery of tyrosine kinase inhibitors at non-tumor sites through nanodrug delivery systems offers further refinement. Real-time in vivo monitoring of CAR T cells using cell tracking techniques with iron oxide nanoparticles detected by MRI and other innovative methods promises enhanced insights into CAR T cell activity. These elements are consolidated in our theoretical model, termed the “addition by subtraction” model (Figure 4). Finally, it is crucial to acknowledge that the clinical application of CAR T nanosymbionts hinges on resolving current questions in nanomedicine regarding technical specifications, cost-effectiveness, scalability, and regulatory considerations.

## Figures and Tables

**Figure 1 ijms-25-13157-f001:**
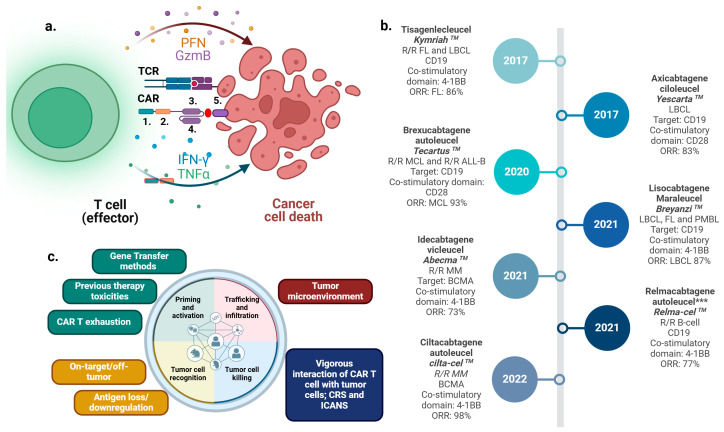
(**a**) CAR T cells work as effector cells recognizing cancer antigens (CD19 or BCMA for FDA-approved CAR T cells) through an immune synapse that contains the following structural elements: 1. a co-stimulatory domain (4-1 BB or CD28), 2. a transmembrane signaling adaptor (CD3ζ), 3. a variable light (VL) chain, 4. a variable heavy (VH) chain, and 5. a tumoral antigen. The interaction between the receptor and its antigen leads to the release of cytotoxic agents such as INF-γ, TNF-α, PFN, and GzmB, leading to cancer cell death. (**b**) A timeline that illustrates the currently approved CAR T cell products, their targets, their co-stimulatory domains, and the overall response rates they achieve. (**c**) Current pitfalls in the biology of CAR T cell therapy. BCMA: B cell maturation antigen; LBCL: diffuse large B cell lymphoma; FL: follicular lymphoma; GzmB: granzyme B; IFNγ: interferon-gamma; LBCL: large B cell lymphoma; MCL: mantle cell lymphoma; MM: multiple myeloma; ORR: objective response rate; PFN: perforin; PMBL: primary mediastinal B cell lymphoma, TNFα: tumor necrosis factor α. *** Approved in China only. Created with BioRender.com.

**Figure 2 ijms-25-13157-f002:**
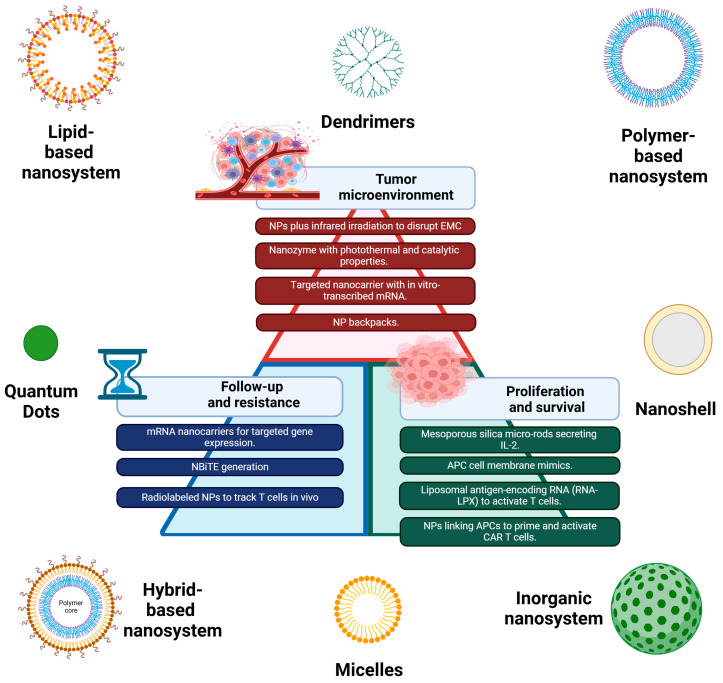
Schematic representation of lipid, polymer, inorganic, and hybrid hydrophobic polymeric nanoparticles (NPs) and the possible advantages of using nanotechnology in CAR T cell therapy: (1) Tumor microenvironment remodeling: using indocyanine green nanoparticles plus infrared light irradiation to disrupt the ECM before CAR administration, using targeted nanocarriers with in vitro transcribed mRNA to reprogram TAMs and downregulate PD-L1, and using nanozymes and nanoparticle backpacks. (2) Improving T cell proliferation and lifespan with mesoporous silica micro-rods secreting IL-2, APC cell-membrane mimics, using RNA-LPX to activate T cells, and NPs linking APCs to prime and activate T cells. (3) Improving follow-up and resistance with genetic programming using mRNA nanocarriers for targeted gene expression and NBiTE generation and radiolabeled NPS to track T cells in vivo. APC: antigen-presenting cell; ECM: extracellular matrix; IL-2: interleukin-2; NBiTEs: nano-bispecific T cell engagers, NPs: nanoparticles, PD-L1: programmed cell death ligand-1; TAMs: tumor-associated macrophages. Created with BioRender.com.

**Figure 3 ijms-25-13157-f003:**
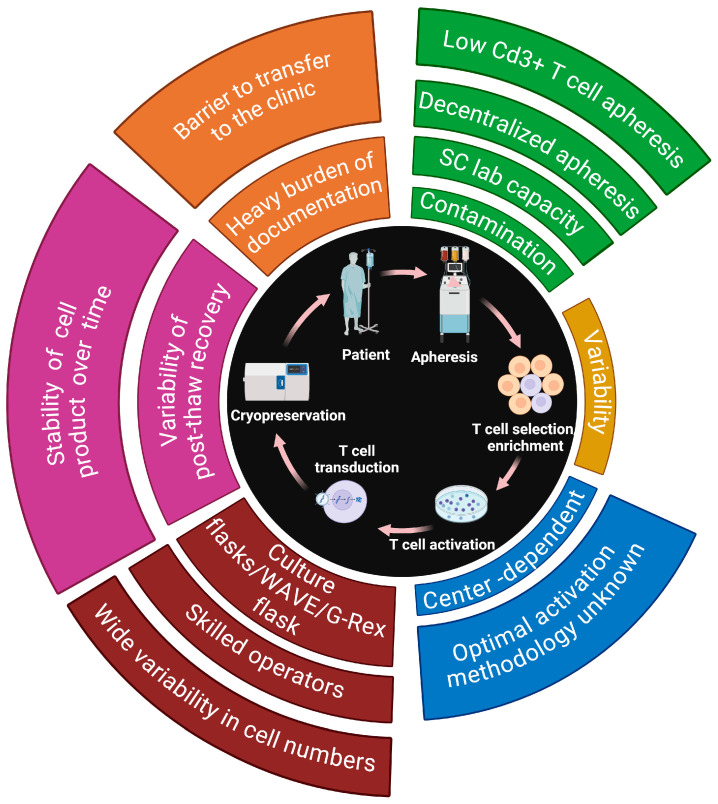
A schematic representation of the issues and challenges that arise with the manufacture of CAR T cell therapies. The black circle encapsulates the current ex vivo process of CAR T cell generation. Adjacent to this circle, we list the prevalent challenges associated with each step. Addressing these challenges might improve the viability of implementing CAR T cell therapies in low-income countries by reducing costs and improving availability. SC: stem cell. Created with BioRender.com.

**Figure 4 ijms-25-13157-f004:**
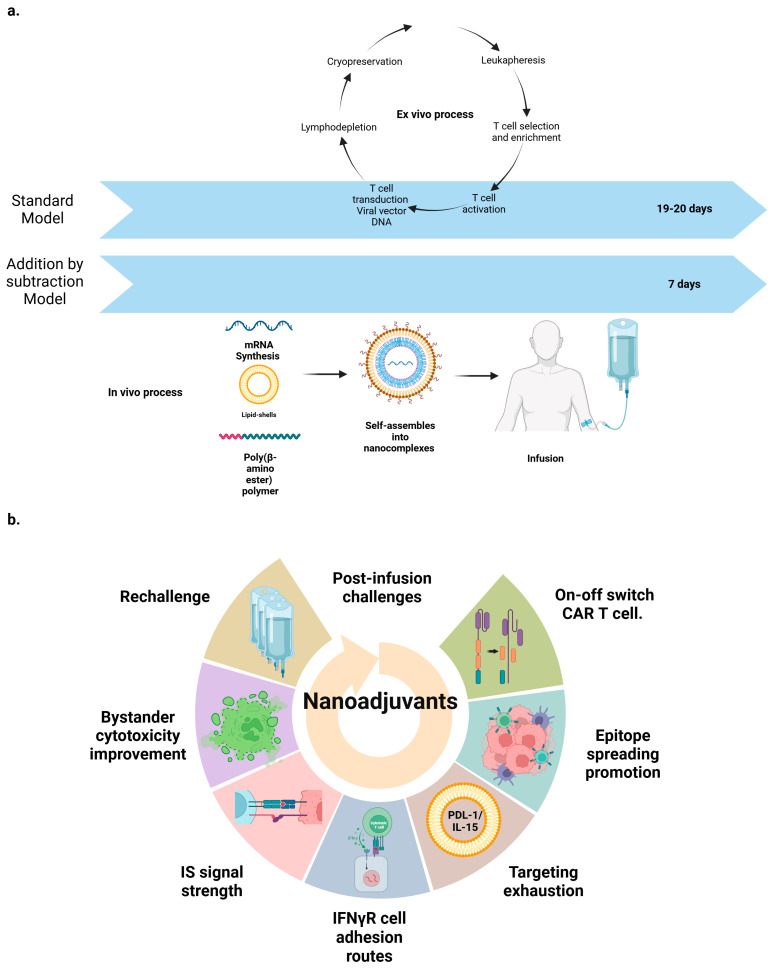
(**a**) The conventional CAR T cell production model presents significant challenges for widespread application. In contrast, the “addition by subtraction” model can reduce production time by using an in vivo approach instead of an ex vivo one; costs can be lowered by decreasing the number of process steps and replacing viral vectors with non-viral vectors (e.g., nanoparticles). The risk of insertional mutagenesis can be better controlled and the toxicity profile balanced by using messenger RNA (though this would simultaneously create the need for multiple therapy infusions). (**b**) Once the CAR T cells are infused, the use of adjuvants with nanotechnology could be considered based on the encountered situation. This could involve modulating CAR T cell activity if significant toxicity is evident, overcoming exhaustion states through enhancement of bystander cytotoxicity, promoting epitope spreading, stimulating alternative lymphocyte activation pathways, and strengthening the IS. IS: immunological synapse. IFNγR: interferon gamma receptor. Created with BioRender.com.

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
