# Peer review of "CAR T Cell Nanosymbionts: Revealing the Boundless Potential of a New Dyad"

_ijms, 2024, doi:10.3390/ijms252313157_

Round 1

Reviewer 1 Report

Comments and Suggestions for Authors

This manuscript proposed the concept of CAR-T Nanosymbionts, exploring the potential and challenges of integrating nanotechnology with CAR-T cell therapy, including the biological challenges and design roadblocks arising from CAR-T cells, and the technical challenges of nanotechnology in clinical applications. But the innovation of the article is limited. A similar idea has already been proposed (Ref. Frontiers in Oncology, 2023, 13, 1288383).

Furthermore, there are several concerns:

1.     It is recommended that part 4 should be simplified. I hold the opinion that the focus of this section is not to describe the difficulties and challenges to nanotechnology, but the solutions proposed by other articles.

2.     The title of the article is "CAR-T nanosymbiont", but part 4 is about the challenges encountered in the CAR-T production process, and some points have no relationship to the nanotechnology.

3.     Although a full and broad summarization is achieved by the authors, expert opinions are few. Please express your opinions, statements and prospects as often as possible.

4.     Please confirm whether the pictures in the article are original? If not the original need to indicate the reference. In addition, it is recommended to add more pictures of the article cited in this article, so that readers can better understand it.

5.     A major problem is the lack of logic in the article,Here are some examples:

i) The title of part 2.1 is the immunological synapse, but line 231-241 are missing the correlation, suggesting move this paragraph to introduction.

ii) Line 497-562 should be advanced to the beginning of part 2.3.

iii) Part 3.4 should not belong to the Part 3, it's better for the article structure to start a new one.

6.     Language issue. Some words are a bit strange, such as “nanosymbiont”, please consider if the “symbiont” appropriate?

7.     Only one reference work is cited in some sections, some parts even do not cite the reference,such as line 311-320, 626-629, please add more references to these sections.

8.     Please check and correct the format of each reference carefully to ensure that it conforms to the journal's specifications.

9.     Minor issues.

i) The article needs to use “[]” symbol to cite the reference, please modify.

ii) There are some errors in the serial number of line 244, 728, 934, 1185.

iii) The Conjunctions in subheadings do not need to be capitalized.

Author Response

Responses

Reviewer #1

  • Comment: although the authors acknowledge the existence of articles that propose the implementation of nanotechnology for CAR T cell development to improve their performance, our edition of the concept consist in a model called “Addition by subtraction” in which we modified some of the steps of CAR T cell manufacturing, implementing nanotechnology and at the same time, other types of tools such as in-vivo manufacturing instead of ex-vivo. The risk of insertional mutagenesis can be better controlled, and the toxicity profile can be balanced by using messenger RNA (though this would simultaneously raise the need for multiple therapy infusions) and the end we add the possibility of administering nanoadjuvants according to the clinical situation (i.e. modulating CAR T-cell activity if significant toxicity is evident, overcoming exhaustion states through enhancement of bystander cytotoxicity, promoting epitope spreading, stimulating alternative lymphocyte activation pathways, and strengthening the IS).
  • Comment #1: Thank you for your comment. We removed and summarized elements of section 4 for better clarity. We also added missing references.
    • Le Saux, A. Aubert-Pou€essel, K.E. Mohamed, P. Martineau, L. Guglielmi, J.-M. Devoisselle, P. Legrand, J. Chopineau, M. Morille, Interest of extracellular vesicles in regards to lipid nanoparticle based systems for intracellular protein delivery, Adv. Drug Deliv. Rev. 176 (2021) 113837
  • Comment # 2: We have revised the section on the CAR-T production process, summarizing it and emphasizing how nanoparticles could improve the overall CAR-T manufacturing process. Additionally, we have attempted to explore more connections between Car T cell and nanotechnology in this specific section. Thank you

  • Comment #3: Expert author comments were added as follow:

Expert comment starting from line 98 to 102

Nanoparticles can carry multiple components simultaneously—such as tumor-associated antigens, adjuvants (immune stimulators), and targeting molecules—allowing for the creation of multifunctional platforms. This combination approach can enhance the immune response in a more coordinated and efficient way, boosting both the innate and adaptive immune systems

Expert comment starting from line 123:

In addition, advanced nanotechnology enables the possibility of developing personalized anticancer vaccines. Tumor antigens can be loaded onto nanoparticles tailored to the individual patient’s tumor profile. These personalized vaccines can trigger a specific immune response against the patient's unique tumor antigens, improving treatment outcomes.

Expert comment starting from line 434

Thus, combination of CAR-T cells and these APC-activating nanoparticle technologies may enhance epitope spreading during CAR T-cell therapy and reduce post-CAR T-cell therapy recurrences due to antigen escape.

Expert comment  at the end of  “The hurdles of living drug tracking” section

Thus, these technologies can reveal in vivo dynamics of both CAR T-cells and nanoparticles, promoting further understanding of CAR T-cell biology in vivo and optimal design of CAR T-cells and nanoparticles.

Expert comment starting from line 1419:

Nanoparticles are an ideal platform for combination therapies, where cancer vaccines can be used alongside other immunotherapies, such as checkpoint inhibitors or CAR T-cell therapies. Nanoparticles can help co-deliver these agents in a synergistic manner, potentially leading to a more robust and comprehensive anticancer response

  • Comment #4: Figures are original, made by the authors for this article only. The software used was BioRender.

  • Comment # 5
  1. We have reorganized the lines from 231 to 241 to better clarify the relationship between immune synapsis and the potential solutions to improve the immune synapse between the CAR-T cell and the tumor cell using nanobodies. This restructuring should now make the connection between the mechanisms involved in immune synapsis and the proposed use of nanobodies more transparent and easier to follow:

“ The immunological synapse, essential for CAR-T cell function, relies on the precise interaction between the CAR's antigen-binding domain and tumor cell epitopes. Currently, this recognition is mediated by the scFv fragment, which can suffer from misfolding, aggregation, and overstimulation of T cells, leading to early exhaustion. Replacing the scFv with a nanobody—small, single-domain antibodies derived from camelid species—offers significant improvements.

Nanobodies, due to their compact size, high stability, and reduced immunogenicity, create a more efficient and stable synapse. Their ability to avoid misfolding and aggregation ensures a more controlled activation of CAR-T cells, preventing the overexpression of cytotoxic signals that could prematurely exhaust the T cells. By enhancing specificity and affinity, nanobodies improve antigen recognition, leading to a stronger and more durable synapse between the CAR-T cells and tumor cells.

Additionally, nanobodies can be engineered into modular structures that facilitate the redirection of universal CAR-T cells to target various tumor antigens, further enhancing the precision and adaptability of the therapy. This combination of stability, specificity, and versatility helps optimize the immune synapse, improving both the efficacy and longevity of CAR-T cell responses against cancer”

  1. Dear Reviewer,

Please note that lines 497 to 562 were placed starting from line 448, meaning at the beginning of section 2.3, immediately following a brief introduction to what the hostile solid tumor microenvironment is.

                        Thank you for your understanding.

  • we have moved section 3.4 to 2.5, now titled "The Hurdles of Living Therapy Tracking," to align it better with the discussion on the biological challenges of CAR T-cells.This change aims to improve the document's flow and thematic coherence.

  • Comment # 6.

Dear reviewer,

I would like to clarify that the term nanosymbionts in our article is actually part of a more complex concept, specifically CAR-T nanosymbionts. The intention behind this terminology is to emphasize the collaborative integration of two powerful technologies—CAR-T cell therapy and nanotechnology. The term "nanosymbionts" is used here to draw an analogy to biological symbiosis, highlighting the mutual cooperation between these two systems. Just as symbiotic organisms work together for mutual benefit, CAR-T nanosymbionts exemplify how these technologies synergize to enhance therapeutic efficacy. This analogy helps to underscore the collaborative nature of the approach we are discussing.

Thank you for considering this clarification, and I hope this helps convey the significance of the term more clearly.

Also we have highlighted some individual words in red throughout the text, which have been corrected to improve the writing, according to the given recommendations.

  • Comment # 7

Dear Reviewer, We have added reference 37 in line 311 and included a new reference as well. Thank you for your feedback.

(Zhu C, Wu Q, Sheng T, Shi J, Shen X, Yu J, Du Y, Sun J, Liang T, He K, Ding Y, Li H, Gu Z, Wang W. Rationally designed approaches to augment CAR-T therapy for solid tumor treatment. Bioact Mater. 2023 Nov 26;33:377-395.)  

Please note that the reference for the information in lines 626 to 629, demonstrating how various nanodelivery systems have been developed to deliver cytokines or regulate cytokine expression in tumor cells, can be found in references 89, 90, and 91.

Comment # 8

We have checked  and corrected  the format of each  reference carefully to ensure that it conforms the journal´s  specfications

Comment # 9

  1. We have modified “[]” symbol to cite the reference
  2. We have corrected the serial number of line 244, 728, 934, 1185

Conjunctions in subheadings were corrected

Reviewer 2 Report

Comments and Suggestions for Authors

Baena et al. provided a detailed overview of the field of CAR-T cell therapies, including their development and pitfalls, mechanisms of action, and clinical applications, as well as an in-depth discussion of nanotechnology-based systems designed to synergistically interact with CAR-T cells to enhance their therapeutic efficacy, targeting capabilities, and safety profile in cancer treatment.

The article showed a major work of sumarizing all the literature. Unfortunately, there are major elements to be modified.

  • The text is really exhaustive, but too long. There are some paragraphs that are not fundamental for the purpose of the review, such as paragraph 3.1.

  • The authors must be separated by a comma.

  • The citation format is inconsistent throughout the article. Please make it homogeneous.

  • The figures are too small, especially the text in them.

  • In the first image, fix the "y" of IFN-γ. The word "CAR" must be written in all capital letters. In the b. panel, it is "Maraleucel"; in the c. panel, in the cell it should be "tumor cell recognition."

  • Please be consistent in the writing of "CAR T cells" or "CAR-T cells" throughout the text. The same applies to "ICAM-1."

  • In line 67, it should be "PFN."

  • Please check for additional spaces before the full stop.

  • Please put a space before the citation in the text.

  • Please check the indentation.

  • Please, if citing a figure, use the full figure name.

  • Write "in vivo," "in vitro," and "ex vivo" in italics and without hyphens.

  • Please cite the software used to create the figures.

  • Check the separation of words at the end of the line for correctness.

  • Check the titles of the paragraphs, as some of them are not in the correct order. Ensure consistency in the titles, such as capitalizing all words and removing full stops at the end.

  • In line 204, remove the full stop before the citation.

  • In line 213, remove the space before the beginning of the sentence.

  • Lines 264 and 265 are a repetition of the concept expressed in the previous sentence.

  • In line 318, confirm if they are referring to PD-1 on T cells. The citation appears to be incorrect.

  • Line 317 should be "FoxP3."

  • Line 341, add a space before the beginning of the sentence.

  • The language is often too colloquial, such as in line 371.

  • Please check the symbols for alpha, beta, and gamma throughout the text, as they are incorrect in some places, such as line 392.

  • Line 445: "nanoparticles" should be one word.

  • Line 537: the concept is unclear.

  • Line 540: this is a repetition; it is already specified that they are derived from macrophage membranes.

  • Lines 562 and 571: it should be "pre-metastatic."

  • Line 585: it should be "pro-apoptotic."

  • Lines 607 to 613: there is no link to the rest of the paragraph or the previous context.

  • Line 622: check the spelling of the polymer.

  • Line 644: the sentence is random and needs a better explanation and references.

  • Line 649: there is no need to explain the etymology of the word.

  • When citing a brand, please include the company reference, such as in line 740.

  • Line 774: it should be "stimulation."

  • There is underlining present in fdifferent sections; please correct it, such as in line 830.

  • Line 842: it should be "transfection."

  • The paragraph titled "The viral vector issue" is too long, and the details sometimes seem not fitting for the purpose of the topic.

  • Gene names must be written in italics, such as TET2 or HMGA2.

  • Please be consistent and ensure there is a space after each paragraph.

  • Line 1080: place the full stop after the citation.

  • Line 1084: the meaning of the sentence is unclear.

  • Line 1117: the citation is missing.

  • Line 1127: insert a hyphen between "spatio" and "temporal."

  • Line 1136: remove the brackets; they can be substituted with commas.

  • Line 1153: check the superscript for the copper isotope.

  • Line 1180: check for underlining.

  • Remove the colon from the title of the paragraph and start a new paragraph for the text.

  • Line 1187: remove the capital letter in "contamination."

  • Paragraph 4.1.3: what is the toxicity of the nanosystem? It has not been previously mentioned, and the overall paragraph must be explained better.

  • Line 1237: a full stop is missing.

  • Line 1298: the meaning is unclear ("invoves using").

  • Line 1306: it should be inserted before.

  • Line 1318: check the sentence, as it is incorrect.

  • Line 1324: use "laboratory" instead of "lab," and if referring to scaling up, add the hyphen.

  • Line 1335: there is a missing bracket.

  • Figure 4: Panel a: The text should be larger. "Ex vivo" must be in italics. It is unclear why the upper process is overimposed on the arrow, and if there is no patient, why the "ex vivo process" is a circle. Use "mRNA" instead of "RNAm," and "in vivo" must be in italics. Panel b: It should be "post-infusion," "CAR" in all capital letters, and "PD-L1." Check the spelling of the words in the panel, specifically "IFN-γ receptor," which is also written incorrectly in the figure description.

Comments on the Quality of English Language

Some of the sentences are overly long, making their meaning difficult to understand.

I recommend having the text reviewed by a native speaker to improve clarity and readability.

Author Response

Reviewer # 2

  1. Dear reviewer. We have summarized paragraph 3.1 and focused on linking it with the solutions offered by Nanotechnology. This revision aims to reduce the article's length while emphasizing only the key points that connect both technologies.

  1. We have already separated the authors with a comma

  1. We have made citation format homogeneous

  1. We have made the requested modifications: We increased the size of the figures and the text within them.We corrected the terms you pointed out, such as 'Interferon Gamma', ensuring the correct spelling. We adjusted 'CAR' to be written in uppercase throughout the document. The term 'maraleucel' has been properly formatted. In panel C, we modified 'Tumor Cell Recognition' within the cell as requested

  1. we have chosen the term "CAR T cells" and have replaced it throughout the entire text, without a hyphen between "T" and "cell”. Also, we have chosen ICAM-1 throughout the entire text.

  1. We have corrected “PFN” in line 67

  1. We have checked for additional spaces before full stop

  1. We have placed a space before each citation in the text

  1. We have reviewed the indentation.

  1. We have used the full figure name to cite our figures

  1. We have written "in vivo," "in vitro," and "ex vivo" in italics and without hyphens

  1. We have cited the software BioRender used to develop  our figures

  1. We have checked  the separation of words at the end of the lines

  1. We have ordered the titles of the paragraphs. We have capitalized all words and removed  full stops at the end

  1. We have removed the full stop before the citation in line  204

  1. We have removed  the space before the beginning of the sentence in line  213

  1. We have deleted  lines 264 and 265  because evidently  are  a repetition of the concept  expressed in the previous sentence:

  The A2aR pathway, which is triggered by increased adenosine levels due to tissue damage and cellular stress, inhibits T cell receptor signaling and IFNγ production through elevated intracellular cyclic AMP adenosine levels that are increased in the tumor microenvironment

  1. We have cited apropiate reference in line 318

Li Z, Xie HY, Nie W. Nano-Engineering Strategies for Tumor-Specific Therapy. ChemMedChem. 2024 May 17;19(10):e202300647. doi: 10.1002/cmdc.202300647.

  1. We have  left  Tregs  in line 317

  1. We have revised the wording in the paragraph on line 341 to improve its flow. Therefore, the space before the start of the sentence no longer applies.

  1. We have  modified  several sections  of the article, aiming to adjust  the writing  style so that it no longer appears overly informal or colloquial

  1. We have checked the symbols for alpha, beta, and gamma throughout the text, including line 375

  1. Throughout the text, nanoparticles is now one word

  1. We have revised the wording of the  paragraph containing line 537, and we hope this change clarifies the concept expressed in that section.

  1. We have revised the wording of the paragraph containing line 540 to eliminate  any repeated concepts

  1. In lines 562 and 571, we are referring to studies at preclinical stages conducted in non-human models

  1. The word in line  586 has been changed  by "pro-apoptotic."

  1. The concept in lines 607 to 613 is intended to emphasize that aerobic glycolysis is essential for the effector function of T cells. However, due to competition with proliferating tumor cells, T cells may be blocked, reducing the production of factors such as interferon-gamma, which leads to hyporesponses and expression of different types of activated lymphocytes

  1. In lines  607 to 613 we would like to propose the use of nanosystems to regulate the metabolic environment within tumor beds, enhancing the response to active cellular therapies. These nanosystems could modulate key metabolic pathways, including glycolysis and mitochondrial biogenesis, to optimize tumor metabolism and improve therapeutic outcomes.

  1. We have checked  the spelling of the polymer

  1. Dear Reviewer, in line 644, we have changed the wording to clarify the concept in this section, and we have organized the references more appropriately

  1. We have deleted the etimology of the word trogocytosis

  1. The Viral Vector Issue section has been shortened and rewritten for better clarity and understanding. Emphasis has been placed on the current challenges that viral vectors pose in CARTICEL therapy. The section concludes by mentioning that many of these difficulties could potentially be overcome by using alternative vectors, such as nanotechnology:

Viral vectors are the preferred method for gene transfer in CAR T cell production due to their efficiency in delivering genetic material to target cells [140]. They are classified into viral (VV) and non-viral (NVV) types, with lentiviruses (LV), derived from HIV-1, being the most common in CAR T therapy. LV integrate genetic material into the host cell by transducing the gene of interest [142]. To produce LV vectors, HEK293T cells are transfected with plasmid DNA (pDNA), leading to the assembly and release of viral particles into the culture medium [143]. This process is cost-effective compared to adeno-associated viruses as it doesn't require a lysis step [144]. LV are genetically modified to increase their genetic payload, reduce pathogenicity, and prevent replication [59], though they face limitations in safety, cost, and flexibility compared to non-viral nano transfer systems.

3.2.1 Safety Concerns: Lentiviral (LV) vectors are designed with separate plasmids to prevent competent viral replication. However, integrating transgenes into the genome can raise the risk of insertional mutagenesis, which varies by vector type [146]. Despite this, genotoxicity remains rare, and LV vectors are considered safer than γ-retroviruses due to their tendency to integrate into transcribed genes [147]. There are no reported cases of oncogenic transformation linked to LV T-cell transduction, but three instances of insertional mutagenesis exist in the literature. Shah et al. reported clonal expansion in a patient treated with anti-CD22 CAR T-cell therapy, resulting in remission [150]. Fraietta et al. found CAR T-cell expansion in a patient receiving anti-CD19 CAR T-cell therapy, linked to LV integration in the TET2 gene [151]. Lastly, Cavazzana-Calvo et al. described clonal expansion in a patient with thalassemia, caused by LV integration into the HMGA2 gene, but the clone eventually dissipated [152]. Some alternatives to prevent genotoxic issues include using transgenes with "suicide genes" or OFF-switches to control CAR T-cell activity without harming unmodified cells [153]. One example is the herpes simplex virus-thymidine kinase (HSV-TK) system [154], which requires prodrugs like ganciclovir for activation but poses challenges such as slow activation and immunogenicity [155,156]. Another approach is using mRNA for CAR expression, which reduces side effects [148] but requires repeated treatments due to its short-lived expression [157].

3.2.2 Transduction efficiencies

Quantification of transfection capability is another challenge to cover when using VV, mainly because there is a variability between lot-to-lot contributing to inconsistency of the therapeutic T cell production, which ranges from 5 to 39% CAR T cells, modifying the number of nucleated cells to deliver 1.7 × 108 to 50 × 108 and subsequent therapy efficacy [158]. On the other hand, non-viral vectors yielded more uniform and consistent CAR expression in >40% of the T cells, showing superior anti-tumor activity [159].

3.2.3 Accessibility

Viral production for clinical applications under Good Manufacturing Practice (GMP) standards takes 2-3 weeks and requires highly regulated processes to ensure product safety and quality. GMP involves the use of biosafety level 3 (BSL3) clean rooms, stringent safety testing, and trained staff to maintain controlled and reproducible conditions [160]. However, a limited number of third-party suppliers monopolize viral vector production, raising concerns about access and equity . FDA guidelines mandate extensive testing to prevent the occurrence of replication-competent viruses during vector production and ex vivo cell therapy product release, with long-term follow-up of up to 15 years [149]. These regulatory requirements increase costs and reduce global availability. In contrast, non-viral gene transfer offers a more efficient alternative, with faster production, lower costs, and no risk of viral replication [60]. CAR T cell therapies, for instance, rely on costly lentiviral vectors, which require custom packaging, strict temperature control, and can cost between $950 and $1250 USD for RNA lentiviral vectors [75]. Non-viral systems could replace these with cheaper and faster methods, improving accessibility.

  1. We have italicized the gene names

  1. We have revised  the text to ensure consistency and have organized the spacing after each paragraph

  1. The full stop after the citation in line 1080 has been added

  1. We have revised the wording of line 1084  to clarify the meaning of the sentence:

Perhaps CAR T nanosymbionts using mRNA can also help to alleviate recent fears concerning safety signal announced by the FDA Center for Biologics Evaluation and Research related to T cell lymphoma development in several patients undergoing CAR T cell therapy for pediatric acute lymphocytic leukemia, non-Hodgkin lymphoma and multiple myeloma with malignant clones containing the genetic signature of the CAR construct

  1. The citation in line 1117 is no longer missing

  1. We have inserted a hyphen  between  "spatio" and "temporal"

  1. We have removed the brackets in line 1136

  1. We have corrected the superscript  for the copper isotope

  1. We have check for underlining in line  1180 and we have removed  the colon from the title of the paragraph and started a new paragraph for the text

  1. We have removed the capital letter in "contamination” in line  1187

  1. We have revised the wording of the  paragraph 4.1.3, and we hope this change clarifies and explain the concept expressed in that section in a better way

  1. Line 1237: a full stop is missing. We have revised  and summarized section 4.1, Technical characteristics of nanosystems, which has also resulted in changes to the puntctuation

  1. We have made modifications and summarized the entire section four, titled “ Technical characteristics  of nanosystems.” With these changes, we hope  the meaning of the paragraph  on line 1298 will be clearer

  1. Line 1306: it should be inserted before. We have made modifications and summarized the entire section four, titled “ Technical characteristics  of nanosystems.” With these changes, we hope problem with line 1306 was resolved

  1. We have made modifications to the content  paragraph in  line 318 to enhance the coherence of the concepts and clarify the sentences.

  1. We have used "laboratory" instead of "lab”. We also have added the hyphen to the words scaling up

  1. We have corrected the missing bracket in line 1335

  1. We have made the following modifications to Figure 4 in response to your comments:
  2. We have extended the text and italicized "ex vivo." The "ex vivo" process is represented as a circle, following the convention used in reference articles.
  3. The arrows in the figure indicate the time required for the processes, from apheresis to the reinfusion of the CARs into the patients.
  4. The term "mRNA" has been modified.
  5. In panel B, we have organized the word "post-infusion," "CAR" in all capital letters, and "PD-L1.". We also have corrected the spelling of the words  in the panel

Reviewer 3 Report

Comments and Suggestions for Authors

The authors present an interesting review of the role of nanoparticles in the treatment of different diseases, using CAR T cells. However, the manuscript needs to be improved before being considered for publication.

1) The order of the sections needs to be changed. Section No. 3 needs to be section No. 2 (after introduction), the authors need to describe what are CAR T cells before moving to explain the contribution of nanotechnology. Section No. 2 needs to be Section No. 3, the molecular contributions and details of nanosystems needs to come after CAR T cell section. The rest of the sections are fine.

2) For the section about Car T Cell description (generation, etc.), the authors do not need to be as descriptive as they are. This technology is a hot topic, and many manuscripts already deal with the general concept of CAR T cells and their generation, etc. I rather recommend reducing this and focusing more on the contribution of nano systems in solving the problems that arise during the production of these Cells.

3) Current Section NO. 2 contains a lot of molecular mechanisms and details that feel dissociated from the manuscript. Although a lot of them are important, I ask the authors to try to reduce all the information that is not completely necessary and focus more on the nanotechnology and how it can help CAR T cells. In some sections, it feels more like a description of molecular mechanisms of T cell activation, etc.

4) Section No. 4 contains technical characteristics of nanosystems as well as challenges related to the production and usage. I think the technical characteristics can be briefly mentioned in the same section as CAR T cell generation or immediately after. The challenges can be left at the end of the manuscript as they are right now.

5) Please adjust the figure order to match the new section order. Also, increase the font size of the figures, they are hard to read. 

6) In general, please remove all unnecessary details that can make the manuscript hard to read and do not contribute to the science that the authors are trying to convey.

Please address these issues before re-submitting the manuscript.

Author Response

Reviewer 3

Comment #1 : Dear Reviewer, we would like to inform you that we have changed the order of sections 2 and 3, placing section 3 before section 2. Additionally, we have summarized section 3 to reduce the length of the article and to focus more on the topics related to nanotechnology and Car T cell therapy. Thank you for your valuable feedback, and we hope these changes meet your expectations

Comment #2: Dear Reviewer, In the section describing CAR T cell therapy, we have made an effort to condense the content, thereby reducing its length. Additionally, we have prioritized highlighting the contributions of nanotechnology in addressing the challenges associated with CAR T cell technology, as suggested. We hope these adjustments meet your expectations.

            “The initial step in CAR T-cell therapy involves collecting peripheral blood mononuclear cells (PBMCs) via leukapheresis. A significant challenge here is the variability in the starting material due to patient-specific factors such as age, disease stage, and prior treatments, which can compromise T-cell quality and function. For instance, patients with non-Hodgkin lymphoma or acute lymphoblastic leukemia often have decreased concentrations of memory T-cells after standard treatments, leading to manufacturing failures or suboptimal CAR T-cell function [104, 105]. Cryopreservation further reduces the viability and recovery of PBMCs compared to fresh products. Additionally, logistical limitations with fresh apheresis products, such as narrow viability windows for manufacturing, complicate the process. Variability introduced by different apheresis reagents, instruments, and personnel can also affect the quality of the collected cells. These challenges highlight a need for technologies that can standardize and preserve T-cell quality during collection and storage [106].

Post-leukapheresis, contaminants like platelets, plasma, and residual anticoagulants can adversely affect T-cell activation and expansion. Current separation methods, such as immunomagnetic separation using antibody-coated beads, are costly, complex, and may introduce additional contaminants or inadvertently activate T cells prematurely, leading to rapid exhaustion or tonic signaling [104,107–109]. There's also a risk of unwanted cell types, like NK cells, contaminating the product. These methods require GMP-grade reagents and rigorous protocols to prevent undesirable effects, adding to the complexity and cost. The need for label-free, efficient, and less cumbersome T-cell enrichment methods presents an opportunity for nanotechnology-based solutions that can selectively isolate T-cell subsets without the drawbacks of current techniques [110].

Effective T-cell activation is crucial for successful CAR T-cell manufacturing [111]. Traditional methods involve using anti-CD3/CD28 antibody-coated paramagnetic beads, which must be meticulously removed before infusion to avoid adverse patient reactions. This bead removal process is labor-intensive, time-consuming, and increases the risk of contamination, requiring multiple operators and careful handling [104,112–116]. Alternative activation methods, such as soluble activation reagents or artificial antigen-presenting cells (APCs), exist but may still have limitations regarding efficiency, scalability, or cost [117]. The variability in activation protocols across different centers leads to inconsistencies in the final CAR T-cell products, affecting clinical outcomes. There is a clear need for innovative activation strategies that are efficient, scalable, and reduce variability—an area where nanotechnology could offer significant advancements [118]. 

Gene delivery is a critical step, traditionally achieved using viral vectors like retroviruses and lentiviruses [117]. While effective, these methods pose safety risks such as insertional mutagenesis and potential oncogenesis due to random integration into the host genome. Viral vector production is also expensive, labor-intensive, and subject to batch variability, leading to inconsistent transduction efficiencies (ranging from 4% to 70%). Non-viral methods, like electroporation of plasmid DNA or transposon/transposase systems (e.g., Sleeping Beauty), offer alternatives but have limitations, including lower efficiency, extended culture times, and potential genomic instability [119–121, 122].. Moreover, CRISPR-Cas systems for gene editing present challenges in delivery efficiency and safety concerns like off-target effects . These issues highlight the necessity for safer, more efficient, and cost-effective gene delivery methods [104].

Researchers are employing gene-carrier nanoparticles to efficiently express chimeric antigen receptors (CARs) in T cells while minimizing toxicity to target cells [124,125]. This innovative approach includes reprogramming T cells directly within the body using nanocarriers, which eliminates the need for external cell manufacturing processes. Studies have demonstrated that CD3-targeted nanoparticles carrying plasmid DNA can deliver leukemia-specific CAR genes to T cells in vivo, resulting in disease remission. Further research using CD3- and CD8-targeted nanocarriers loaded with in vitro transcribed mRNA has achieved effective CAR expression in T cells, leading to disease regression in mouse models of leukemia, prostate cancer, and hepatitis B-induced liver cancer [126].

Another study focused on creating antifibrotic CAR T cells that target fibroblast activation protein (FAP), a marker of fibroblast activation. Delivering FAP-CAR-mRNA via CD5-targeted lipid nanoparticles reduced fibrosis and improved heart function in rodent models with tissue scarring. These nanoparticle-based systems offer enhanced manageability and precise timing in clinical applications, positioning them as promising off-the-shelf solutions for various medical conditions [127].

Despite these advancements, concerns about long-term genomic safety persist, including issues like genomic insertion and promoter dependency. While nanoparticles provide comparable treatment results to traditional viral vectors and offer benefits like simplified storage and reduced costs, these safety issues need thorough investigation [128].

Cationic polymers such as polyethyleneimine (PEI) and poly(2-dimethylaminoethyl methacrylate) (pDMAEMA) are used to form complexes with nucleic acids, facilitating their entry into cells by crossing the cell membrane [129]. Nanoparticles typically deliver their genetic cargo via endocytosis or membrane fusion, attaching to sulfated proteoglycans on T-cell membranes. Nanoparticle-sensitized photoporation is another efficient method that opens pores on cell surfaces to allow external cargo entry, capable of producing engineered T cells at a high throughput exceeding 10^5 cells per second.

Lipid nanoparticles are commonly used for nucleic acid delivery because they can encapsulate large nucleic acid molecules and protect them from degradation. Lipidoids, a subtype of lipid nanoparticles, are easily prepared and share many properties with lipids. Combining nanomaterials with electroporation techniques may enhance the transfection efficiency of large DNA plasmids into human primary T cells. A novel, stimuli-sensitive cationic nanomicelle based on a specific block copolymer has shown efficiency in delivering and releasing DNA to targeted sites [130].

Overall, while nanoparticle-based methods show significant promise for producing CAR T cells and treating various diseases, further research is necessary to fully assess their safety and effectiveness, especially for in vivo applications [131,132].

Scaling up CAR T-cell production requires expanding transduced cells to clinically relevant numbers. Traditional static cultures are impractical due to labor intensity and high contamination risk [102,104]. Bioreactors like the G-Rex system and rocking-platform bioreactors (e.g., GE WAVE) offer improvements but come with their own challenges [133–135]. The G-Rex system can disturb cell cultures during sampling, affecting expansion kinetics, while rocking-platform bioreactors are susceptible to mechanical failures and contamination risks due to semi-automated operations [133,134,136–138]. Fully automated systems like CliniMACS Prodigy aim to streamline the process but are limited by their inability to process multiple batches simultaneously, potentially slowing down production [133]. These limitations point to a need for advanced, fully automated, and scalable expansion technologies that ensure consistent cell quality and reduce contamination risk. Nanotechnology could contribute to developing microfluidic bioreactors or nanoscale scaffolds that provide a controlled environment for efficient T-cell expansion [139].

Before infusion, CAR T-cell products undergo cryopreservation and extensive quality control testing to meet GMP standards. However, studies show that a notable percentage of manufactured products fail to meet release criteria due to variability in cell numbers and quality. The GMP process also imposes a heavy documentation burden and navigational challenges within a varied regulatory landscape. These complexities can delay treatment and increase costs. There's a pressing need for technologies that simplify quality control processes and enhance the consistency and stability of the final CAR T-cell product. Nanotechnology may offer advanced biosensing and analytical tools for more efficient quality assessments. [106].”

Comment #3:

  1. Dear Reviewer, we have begun modifying section 2 in an effort to reduce the detailed molecular mechanisms that might detract from the central focus of the manuscript. Specifically, we started by condensing the information in subsection 2.1 concerning the immunological synapse, (from line 165 to line 185) to better emphasize the contribution of nanotechnology in this specific context of CAR therapy.

Thank you for your feedback.

“The immune synapse (IS) is crucial for CAR T cell activation, triggering cytotoxic lymphocytes (CTLs). Unlike conventional T cells, CAR T cells form a disorganized IS configuration, using Lck/ZAP70 signaling to rapidly establish the CAR T cell IS (carIS) within 5 minutes. This leads to intense signaling, quick detachment from target cells, and a "serial killer" pattern of cancer cell destruction. CarIS shows higher expression of molecules like Bcl2 and PEA-15, which have antiapoptotic and antiproliferative properties. Enhancing Fas-mediated apoptosis through agents like histone deacetylase inhibitors (HDAC inhibitors or HDACi) can boost CAR T cell effects, but systemic HDACi use has challenges like poor pharmacokinetics, low specificity, and drug resistance.”

  1. Dear Reviewer, in the article we originally submitted for review, we have removed a couple of lines from line 209 to line 219 to simplify the concept of antigen density in the immune synapse.

“The effectiveness of CAR T cells depends on high tumor antigen density, but this is often variable. Modifiable factors like co-stimulatory domains influence CAR T performance. Enhancing signal strength can involve altering co-stimulatory domains, adding activation motifs, using chimeric receptors, or modifying the hinge-transmembrane region. Proper selection of these domains is key to improving CAR T cell therapies”.

  1. We have carefully considered your feedback and made revisions to the introductory section of the manuscript "Beyond Binding of Immune Synapses: A Multi-Angle View." Specifically, starting at line 246 of the original manuscript, we have summarized this section to reduce the overall length and to place greater emphasis on the advantages of utilizing nanotechnology in CAR-T cell therapy. The revised version now provides a more focused discussion on this particular point, aligning with the core theme of the paper.

While tumor-associated antigens (TAA) are key to CAR T cell function, identifying alternative specific tumor antigens is difficult. Even when a candidate is found, many antigen-positive cells do not respond to CAR T cell reinfusion, implying multiple pathways are involved in immune synapse function and CAR T cell cytotoxicity. In solid tumors, interferon-γ receptor (IFNγR) signaling is critical for cell adhesion after CAR T cell treatment. Disruptions in this signaling can impair immune synapse formation, reduce CAR T cell binding, and lead to resistance. This has been noted in glioblastoma, ovarian, lung, and pancreatic cancers, where ICAM1 expression is reduced when IFNγR signaling is impaired. Conversely, soluble IFNγ can increase ICAM-1 expression but may also trigger cytokine toxicity depending on the tumor type.

  1. Dear Reviewer, we have summarized the introduction of the paragraph on CAR T-cell exhaustion that begins on line 303. Thank you for your valuable feedback.

T cell exhaustion occurs due to increased inhibitory signals from molecules like PD-1, Tim-3, LAG-3, VISTA, CTLA-4, and TIGIT, affecting both tumor cells and T lymphocytes. It is common in over-differentiated CAR T cells, linking a naïve T cell phenotype to better function and clinical outcomes. Factors such as prior chemotherapy and conditioning regimens like lymphodepletion contribute to this exhaustion. Additionally, systemic use of immune checkpoint inhibitors may increase toxicity without effectively targeting the tumor.

  1. We have made an effort to summarize the concept proposed by Wei Dong Nie and colleagues in line 326 to enhance its clarity and comprehensibility:

Nanocarriers, developed by Weidong Nie and colleagues, can effectively deliver PD-L1 antibodies to target exhausted T cells. These magnetic nanoclusters, equipped with PD-1 antibodies, utilize a pH-sensitive bond for attachment and bind to effector T cells through PD-1 receptors. In an acidic environment, they release anti PD-L1 antibodies, blocking PD-1 interactions and maintaining CTL functionality above 90% while delaying tumor progression by over 14 days. The treatment also reduced Tregs and increased CD8+ CTLs in tumor-bearing mice. When exposed to a magnetic field, these nanoclusters enhanced CTL retention at tumor sites, demonstrating their potential for targeted CTL therapy with minimal impact on physiological parameters, indicating safety.

  1. We have also attempted to summarize the concepts from line 340 to line 367 in order to make the message clearer and more concise

Magnetic nanovehicles for targeted drug delivery are designed to enhance the precision and efficacy of cancer treatments. These nanovehicles encapsulate therapeutic agents that are directed to tumors using external magnetic fields, releasing the drug in a controlled manner in response to stimuli such as pH changes, temperature, or enzymatic activity. They also possess theranostic capabilities, acting as contrast agents for magnetic resonance imaging (MRI) to enable real-time monitoring of drug delivery. To overcome T cell exhaustion, chaperone cells are used to direct charged nanoparticles to hard-to-reach anatomical compartments. The conjugation of liposomes and synthetic nanoparticles with CD8+ T lymphocytes via maleimide-thiol coupling provides continuous pseudo-autocrine stimulation of transferred cells. In models of B16F10 melanoma cultures and metastases in the lung and bone marrow, T cells conjugated with nanoparticles showed a 176-fold more efficient accumulation in target tissues compared to intravenous nanoparticles, without increasing toxicity or autoimmunity. Additionally, a multilamellar lipid nanoparticle core loaded with IL-15 and IL-21 released cytokines in very low doses over seven days, resulting in significantly higher proliferation compared to systemic infusion. In murine models, all animals treated with nanoparticle-decorated T cells achieved complete tumor clearance and had longer survival compared to those receiving systemic treatment.

  1. We have summarized the introduction of the section titled "Antigen Escape and Weakness Transgression" from line 371 to line 396 in the clearest way possible to enhance the flow of the concept and emphasize the relevance of using nanoparticles in this specific context:

Reduced expression of targeted antigens and the emergence of antigen-negative cell populations account for 9% to 25% of recurrences after CAR-T cell therapy, due to acquired genetic instability and immunoselection in tumor cells. Despite lymphodepletion pretreatment, residual dying cells are captured by antigen-presenting cells (APCs), which create new MHC Class I and II peptides that prime native T cells to attack tumor cells. This process, known as epitope spreading, involves antigens recognized by these lymphocytes that differ from those initially targeted by CAR-T cells. For effective activation, tumor cryptic antigens must be presented on MHC Class I molecules, engaging cytotoxic CD8+ T cell responses via cross-presentation, primarily performed by specific APCs such as human BDCA3+, XCR1+, and CD141+ dendritic cells. However, the clinical implications of this mechanism and its therapeutic efficacy are not yet fully understood. Some researchers attribute the challenge to inadequate activation of Baft3-dependent dendritic cells, leading to trials of STING agonists in mice to enhance IFN-b responses and improve antigen cross-presentation. In murine models, CAR-T cells combined with repetitive STING agonist injections (2′3′-cGAMP) demonstrated strong synergistic effects, achieving a 50% cure rate and reducing contralateral tumor progression. These results largely depended on CD103+ dendritic cells. However, limitations such as intravenous access, uneven drug diffusion, and difficulty controlling drug distribution hinder the effectiveness and half-life of STING therapies.

  1. We have attempted to summarize the introduction of the section titled "Hostile Solid Tumor Microenvironment" from lines 438 to 453 to emphasize the relevance of using nanoparticles in this specific context.

The tumor microenvironment significantly influences the efficacy of CAR T cell therapy, often leading to T cell dysfunction and therapy failure. Stromal cells, such as cancer associated fibroblasts activated by tumor growth factor b (TGF-β) produce extracellular matrix proteins that inhibit T cell motility. Furthermore, tumor angiogenesis hampers CAR T cell extravasation into the tumor microenvironment, and disrupts adhesion molecules function, decreasing the effectiveness of the immune synapse. High levels of myeloid derived suppressor cells are related with poor prognosis when CAR T cell therapy for hematological and solid malignancies is used. Addressing these challenges, targeting specific pleiotropic cytokines like TGF-b can improve CAR T cell survival. For example, microenviroment derived TGF-β inhibits activation, proliferation and function of cytotoxic T lymphocytes by up-regulation of the regulatory gene FoxP3 and by decreasing the production of IFN-γ, perforin, granzymes A and B, and Fas ligand. At the same time, this molecule impairs natural killer function and benefits regulatory T lymphocytes survival.Systemic efforts targeting TGF-β ligands or receptors in clinical trials have showed discrete results and faced several challenges in terms of toxicity and autoimmunity. For this reason, the concept of nano backpacks is relevant because it utilizes T lymphocytes as vehicles to take loaded nano particles to tumor microenvironment

  1. We have attempted to summarize the concept proposed by Li Tang and colleagues regarding the nano-backpacks, as it appears from line 463 to line 475.

Li Tang and colleagues investigated a "backpacking" method that chemically links an interleukin 15 superagonist to T cells to enhance their activation in melanoma and glioblastoma mouse models. They developed drug-releasing protein nanogels attached to CD45, which activate upon antigen recognition in the tumor microenvironment. This approach led to a 16-fold increase in T cell expansion within tumors compared to systemic cytokine administration, and a 1000-fold increase compared to T cells without cytokine support. Backpacked T cells proliferated and produced effector cytokines in tumors while remaining inactive in circulation, minimizing toxicity. Overall, this strategy effectively delayed tumor growth. (See figure 2)

  1. We have attempted to summarize the concept from lines 471 to 495 to emphasize the relevance of using nanoparticles in this specific context.

Effective interaction between CAR-T cells and cancer cells is crucial for adoptive cell therapy. Solid tumors create physical barriers like dense tissue and compressed vessels, limiting CAR-T cell penetration. To address this, nanophotosensitizer-engineered CAR-T biohybrids (CT-INPs) were developed, using indocyanine green nanoparticles (INPs) attached to CAR-T cells. Upon near-infrared (NIR) laser treatment, these biohybrids induced mild photothermal effects without affecting CAR-T cell function or viability. Tumor cells were destroyed at temperatures above 43°C, resulting in 98% tumor cell death. In a mouse model, CT-INPs plus laser treatment reduced tumor growth for up to 40 days. Ultrasound imaging revealed enhanced blood flow and tumor vessel dilation, improving immune cell infiltration and antitumor cytokine expression. The treatment was well tolerated in mice.

  1. We have summarized from lines 504 to line 519 in the clearest way possible to enhance the flow of the concept

Drug delivery nanosystems transport treatments to tumors, acquiring a biological identity in vivo (like forming a protein corona). Their size, shape, and surface chemistry influence their movement within the tumor microenvironment, affecting cellular targeting. Passive transport through the EPR effect helps nanoparticles accumulate in tumors, with a size cutoff of 200-1200 nm. However, many are trapped by the extracellular matrix, limiting deep penetration. Smaller nanoparticles (< 30 nm) diffuse more easily, while larger ones are often absorbed by tumor macrophages. Nanoparticle size also affects receptor internalization, signaling, and toxicity.

  1. We have summarized from lines 514 to line 550 in the clearest way possible to enhance the flow of the concept

The development of nanorobots for cancer treatment emphasizes the biocompatibility of materials to ensure functionality within tumor tissues. DNA origami represents a major breakthrough in nanorobotics, while viral capsids, which protect viral nucleic acids and can release them upon binding to specific biomarkers, offer a robust natural design for drug delivery. Materials like chitosan, gelatin, alginate, pectin, and dextran have been widely applied in cancer therapies for nanoparticle production. A notable innovation is the multi-component magnetic nanorobot, constructed from multi-walled carbon nanotubes (CNTs) loaded with doxorubicin (DOX) and anticancer antibodies. This nanorobot, driven by an external magnetic field, releases the drug payload in response to intracellular H2O2 or pH changes, particularly within human colorectal carcinoma cells (HCT116).

Other innovations include pine pollen-based magnetic microrobots, which deliver DOX inside cancer cells using magnetic rotors, and ultrasound-driven nanowire motors that achieve rapid, near-infrared light-triggered drug release. A tubular, multi-layer nanorobot combines bubble propulsion with magnetic field guidance to efficiently deliver drugs at high speeds, while porous metal rod-like nanorobots carry significantly larger amounts of drugs than their planar counterparts. Janus mesoporous silica nanomotors, cloaked in macrophage cell membranes, allow for immune-selective binding to cancer cells, enhancing targeted therapy. These advancements in nanotechnology offer potent active drug delivery mechanisms, far surpassing traditional passive systems, and have significant potential for improving cancer treatment.

  1. We have condensed and summarized the "Seed and Soil Hypothesis at the nanoscale" section, spanning lines 574 to 602, with the intention of making the text more fluid and concise for better readability

Tumor-derived exosomes, which are extracellular vesicles (40-160 nm) with a lipid bilayer membrane, act as key immune regulators in both the tumor microenvironment and premetastatic niches. Their internal or surface bioactive cargo, including proteins, lipids, microRNAs, mRNAs, long non-coding RNAs, and DNA, can disrupt normal immune functions. They decrease dendritic cells, inhibit CD8+ T cells, promote the differentiation of immature T cells into regulatory T cells (Tregs), polarize macrophages to an M2 phenotype, suppress NK cell activity, and induce myeloid-derived suppressor cells. These effects help the tumor evade immune detection and prepare distant sites for metastasis. Exosomes can also trigger T cell apoptosis through FasL expression.In preclinical models, tumor-derived exosomes have been shown to impair CAR T cell function by increasing inhibitory receptors like CTLA4 and TIM-3, hindering antigen-specific CAR T cell proliferation and

cytotoxicity. This leads to T cell exhaustion, reflected by downregulation of activation proteins and increased expression of exhaustion markers. Exosomes also carry high levels of PD-L1, which further inhibits CAR T cells, reducing their production of granzyme B and IFN-γ. However, blocking exosome activity has been found to enhance CAR T cell effectiveness.

On the other hand, CAR T cell-derived exosomes offer therapeutic benefits due to their expression of pro-apoptotic and CAR molecules, giving them cytotoxic properties. Their limited lifespan, lack of replication capacity, and ability to cross tumor barriers, combined with low immunogenicity, make them promising tools for improving adoptive cell therapy by reducing side effects and increasing efficacy.

  1. We have summarized the introduction of the section titled "Metabolism and Immune Exclusion," covering lines 605 to 626, to emphasize the relevance of nanoparticle usage in the context of immunometabolism.

Tumor cells preferentially use aerobic glycolysis for ATP production, converting glucose to lactate despite oxygen availability, which supports their synthesis of key biomolecules and enhances growth. CAR T cells and T lymphocytes also rely on this metabolic pathway, leading to competition for glucose in the tumor microenvironment. Aerobic glycolysis is essential for T cell effector function, particularly for translating IFN-g mRNA. However, the glucose-restrictive environment caused by rapidly proliferating tumor cells can metabolically block T cells, reducing IFN-g production and leading to decreased proinflammatory cytokines and T cell hyporesponsiveness over time. Other molecules, such as arginine, tryptophan, and reactive oxygen species, further disrupt T cell metabolism and contribute to immune evasion. Activated T cells differentiate into various subsets: effector T cells (Teff), stem cell memory T cells (Tscm), central memory T cells (Tcm), effector memory T cells (Tem), and tissue resident memory T cells (Trm), each with distinct capabilities. Less differentiated phenotypes, such as stem cell memory and central memory T cells, which have greater potential for self-renewal and resistance to exhaustion, are associated with better clinical responses in CAR T cell therapies.

  1. We have summarized the concepts included from line 628 to line 660 to enhance the readability of the text and clarify the message.

Researchers are striving to balance the killing capacity, expansion, and persistence of CAR T cells. Cytokines such as IL-7, IL-15, and IL-21 promote oxidative phosphorylation, aiding the expansion of Tscm-like and Tcm-like cells, which are linked to long-lasting anti-tumor activity due to their stemness and persistence. Nano-delivery systems have emerged to efficiently transport cytokines or regulate their expression in tumor cells, lowering the required dose and reducing adverse effects while protecting the cytokines from degradation before reaching their target. For instance, poly-γ-glutamic acid-based platforms with chitosan have enhanced cytokine secretion by

macrophages, increasing IL-6, IL-12, and TNF-α levels, which inhibit tumor cell invasion. Additionally, β-cyclodextrin-based nano-systems and adenovirus vectors carrying the IL-12 gene have been shown to inhibit tumor growth in mouse melanoma models. Liposome-based nanomaterials, valued for their histocompatibility and modifiability, have also been used to deliver cytokines, such as mRNA encoding IL-12 and IL-27, to modulate the tumor microenvironment (TME), boosting IFN-γ and TNF-α levels and activating NK cells and cytotoxic T lymphocytes (CTLs). Similarly, inorganic nanomaterials, including mesoporous silica nanoparticles, magnetic nanoparticles, and gold-based structures, have been explored as potential delivery systems.

Additionally, anti-PD-L1 therapy has been found to reduce tumor cell expression of glycolysis enzymes and mTOR protein phosphorylation, increasing glucose availability for T cells, independent of PD-1 expression. The binding of PD-L1 to PD-1 inhibits glycolytic activity and mitochondrial biogenesis, reducing glucose for infiltrating T cells. Nanomachines further enhance local concentrations of anti-PD-L1 molecules, supporting the use of CAR T cell nanosymbionts to regulate the metabolic environment in adoptive cell therapy.

Comment # 4

Dear Author, in Section 4, our focus is not on the technical characteristics of the nanoparticles themselves, but rather on the issues that arise from some of these characteristics. We have attempted to summarize this section and have changed the title to ensure that this focus is highlighted and better understood.

Comment # 5

Thank you very much for your valuable feedback. We wanted to let you know that we have added a second reference on line 281, addressing your observation regarding the limited number of references in certain sections. We greatly appreciate your time and suggestions.

Liu, Q., Zhang, W., Chen, S. et al. SELEX tool: a novel and convenient gel-based diffusion method for monitoring of aptamer-target binding. J Biol Eng 14, 1 (2020). https://doi.org/10.1186/s13036-019-

Comment # 6

We have adjusted the order  of the figures  to match  the revised order of the sections. Additionally, we have  increased  the size of the figures to improve  readability

Comment # 7

We have made an effort to revise all sections of the article  to remove unnecessary details and make the reading experience easier and more user-friendly

Round 2

Reviewer 1 Report

Comments and Suggestions for Authors

This revised version can be acceptable.

Author Response

No corrections sugested

Reviewer 2 Report

Comments and Suggestions for Authors

I appreciate the effort that has been made to improve the manuscript. However, there are still several issues that need to be addressed for further refinement.

  • Check the citations: Ensure all citation numbers are present and correctly ordered in the text. Additionally, make sure the citation style is consistent throughout.

  • Between lines 125-129, insert a citation to support the information provided.

  • In lines 133 and 630, remove the extra spaces.

  • In lines 168, 170, and 177, consider using "CAR-IS" in uppercase, as it is an acronym.

  • There is a missing space in line 242 before the full stop at the beginning of the sentence. Also, in lines 254 and 255, ensure there is a space before the citation.

  • The sentence in line 288 is a repetition of the sentence in line 290.

  • In lines 337 and 373, "naive" should be used instead of "native."

  • In line 242, add the missing full stop at the end of the sentence.

  • In lines 345 and 401, verify that the correct symbol for beta (β) is used.

  • In line 350, the phrase "difficulty controlling drug" is unclear and requires clarification.

  • There are extra spaces in line 373.

  • In line 409, remove the extra space between "nano" and "particles," and add a missing full stop.

  • In line 425, remove the extra full stop.

  • In lines 428, 430, 431, 771, and 772, "CAR-T" should be written as "CAR T" for consistency with the rest of the article.

  • In line 441, the acronym "EPR" is used without prior explanation. Please define it on first use.

  • In lines 454 and 457, add a space after the full stop.

  • In line 507, check that the correct symbol for gamma (γ) is used.

  • In line 517, add the missing full stop.

  • In line 544, remove the extra space after "trogocytosis."

  • In line 542, remove the extra space before "strategies."

  • In line 606, add the missing full stop.

  • In line 754, the use of "they" is unclear, as the previous sentence refers to viral vectors. Please revise to clearly distinguish between viral and non-viral methods.

  • In line 764, do not place a colon after the title.

  • In line 846, fix the brackets and remove the extra full stop.

  • In line 995, fix the brackets.
